# Phosphorylation toggles the SARS-CoV-2 nucleocapsid protein between two membrane-associated condensate states

Bruna Favetta[1], Huan Wang [2], Jasmine Cubuk [3], Arjun Singh[4], Mayur Barai [4], Cesar Ramirez[1], Haiyan Zheng [5], Adam J. Gormley [1], N. Sanjeeva Murthy [2], Gregory Dignon[4], Andrea Soranno [3], Zheng Shi [2] & Benjamin S. Schuster [4] ✉

The Nucleocapsid protein (N) of SARS-CoV-2 plays a critical role in the viral lifecycle by regulating RNA replication and by packaging the viral genome. N and RNA phase separate to form condensates that may be important for these functions. Both functions occur at membrane surfaces, but how N toggles between these two membrane-associated functional states is unclear. Here, we reveal that phosphorylation switches how N condensates interact with membranes, in part by modulating condensate material properties. Our studies also show that phosphorylation alters N's interaction with viral membrane proteins. We gain mechanistic insight through structural analysis and molecular simulations, which suggest phosphorylation induces a conformational change in N that softens condensate material properties. Together, our findings identify membrane association as a key feature of N condensates and provide mechanistic insights into the regulatory role of phosphorylation. Understanding this mechanism suggests potential therapeutic targets for COVID infection.

The COVID-19 pandemic has focused attention on the mechanisms of SARS-CoV-2 viral replication. SARS-CoV-2 is an enveloped virus with a non-segmented, positive-sense, single-stranded, ~30 kb RNA genome[1]. In the virus core, genomic RNA (gRNA) is associated with Nucleocapsid protein (N), forming a ribonucleoprotein (RNP) complex. N has several functions during the viral lifecycle, but is primarily involved in protecting the viral RNA genome by binding, condensing, and packaging it within the virion[2]. N has also been shown to be necessary for efficient transcription and replication of viral RNA, and it additionally contributes to immune evasion via sequestering stress granule proteins[3]. How N is regulated throughout the viral lifecycle to perform its varied functions, and whether it is possible to therapeutically manipulate this regulation to inhibit viral replication, remains unknown.

N is a multidomain protein, consisting of two folded and three disordered domains (Fig. 1a), the diversity of which likely contributes to its ability to perform several functions throughout the viral lifecycle[2]. The N-terminal folded domain (NTD) strongly binds with specific viral RNA elements, stabilizing the RNP complex[4,5]. The C-terminal folded domain (CTD) mediates dimerization of N, facilitating the formation of the helical nucleocapsid structure that protects the viral RNA genome[6,7]. Both folded domains are flanked by disordered regions. The central disordered region acts as a linker that is partially extended and thus retains some structure while providing the protein conformational flexibility[8,9]. The serine/arginine (SR)-rich region (aa 175–206; Fig. 1b) within the linker is known to participate in both protein–protein and protein–RNA interactions[10–12].

[1]Department of Biomedical Engineering, Rutgers, The State University of New Jersey, Piscataway, NJ, USA. [2]Department of Chemistry and Chemical Biology, Rutgers, The State University of New Jersey, Piscataway, NJ, USA. [3]Department of Biochemistry and Molecular Biophysics, Washington University in St Louis, St. Louis, MO, USA. [4]Department of Chemical and Biochemical Engineering, Rutgers, The State University of New Jersey, Piscataway, NJ, USA. [5]Center for Advanced Biotechnology and Medicine, Rutgers, The State University of New Jersey, Piscataway, NJ, USA. ✉e-mail: benjamin.schuster@rutgers.edu

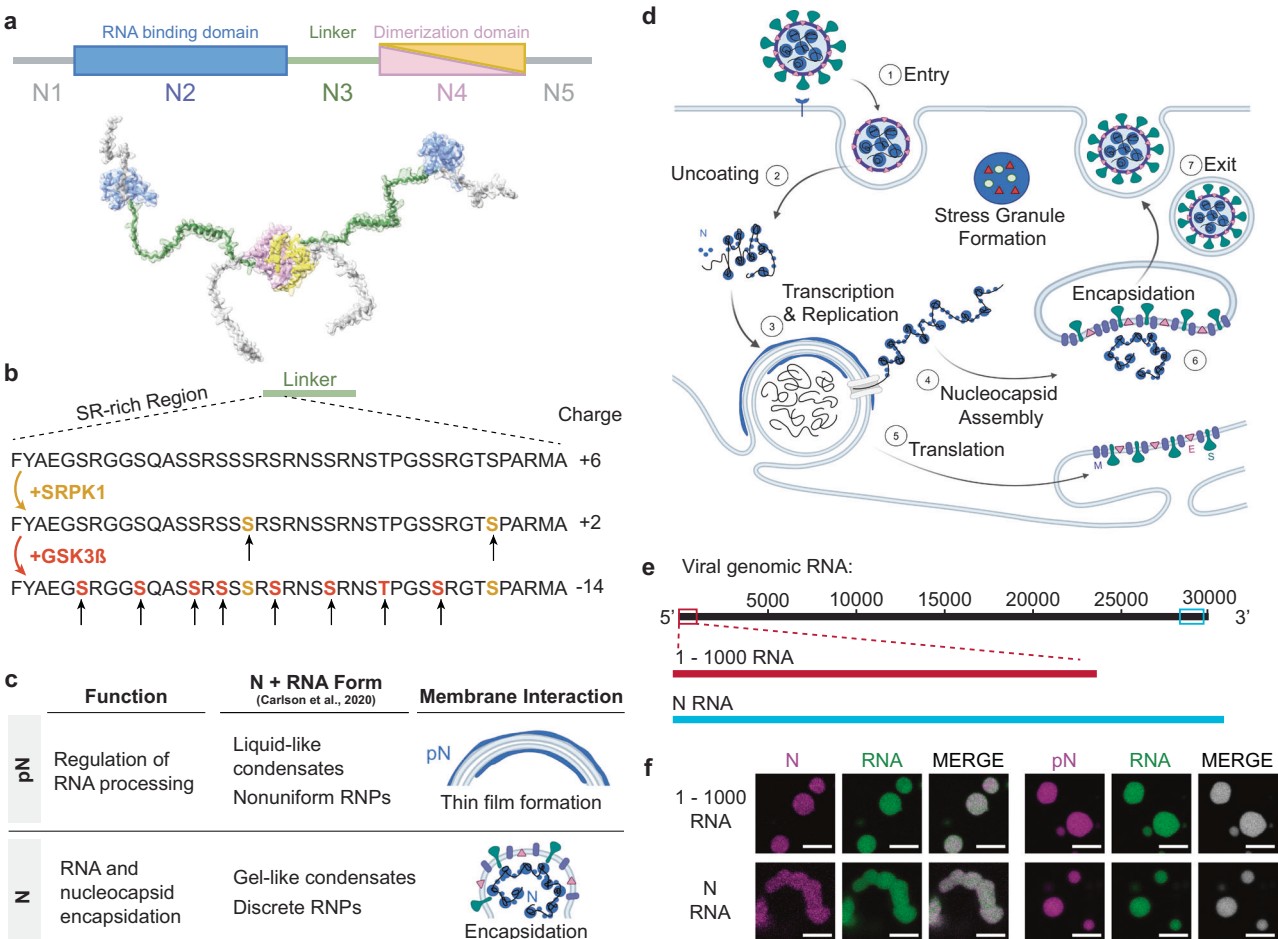

**Fig. 1 | Phosphorylation of the SARS-CoV-2 Nucleocapsid protein (N).**
**a** Schematic of the full-length SARS-CoV-2 Nucleocapsid protein (N) used in experiments, showing N1, a disordered domain; N2, RNA-binding domain (PDB ID: 7VBD); N3, disordered linker domain containing the serine/arginine (SR)-rich region; N4, dimerization domain (PDB ID: 6WZQ); and N5, a disordered domain. **b** Sites of phosphorylation in the SR region of the N3 domain. Our experiments used two kinases—SRPK1 and GSK-3β—resulting in a maximum of 10 phosphorylation sites. **c** Hypothesized form and roles of N and phosphorylated N (pN). pN may have dynamic functions, form liquid-like droplets with RNA, and may localize to the surface of replication organelles. Unmodified N forms solid-like, spherical assemblies with genomic RNA and facilitates new virus encapsulation. N's form with RNA was characterized by Carlson et al., 2020, using light and electron microscopy.

Schematic created in BioRender. Favetta, B. (2025) https://BioRender.com/speaa8j. **d** A diagram of the SARS-CoV-2 lifecycle. Following viral entry (1), the genome is uncoated (2) such that it can be read. Viral RNA is stored within viral replication organelles (3), wherein RNA transcription and replication may be occurring. N binds to and condenses genomic RNA exiting these organelles (4). In parallel, viral structural proteins are produced (5), in preparation for viral encapsulation (6) and exit (7). Schematic created in BioRender. Favetta, B. (2025) https://BioRender.com/o21g999. **e** Main RNA fragments used in experiments are the first 1000 bases from the 5' end of the viral genome and the 1340-base fragment encoding N. **f** Representative images of droplets with varied morphologies form upon mixing 40 μM N (5% Alexa Fluor 647 labeled) and 300 nM RNA fragments (5% Cy3 labeled) at 37 °C from $n = 3$ independent trials. Scale bar = 5 μm.

Importantly, this SR-rich region was identified as a site of phosphorylation of N (Fig. 1b) that may regulate protein function, based on several lines of evidence. (1) At different stages of the viral lifecycle, N exists in two phosphorylation states. Abundant phosphorylated protein is found inside infected cells, while unmodified protein is found within virions[13–18]. Regulation of phosphorylation of N may act as a timer in the viral lifecycle, switching from replication, transcription, and translation to assembly of new virions[19–21]. (2) Phosphorylation of the SR-rich region of N has been shown to modify how the protein interacts with RNA, as well as affect the transcription and translation of RNA[22,23]. (3) During viral replication, when N is likely phosphorylated, it appears to form a thin layer around viral replication organelles (vROs), double membrane vesicles filled with viral RNA commonly found in infected cells[16,24]. However, during virion assembly, complexes of unmodified N and RNA remain as small, spherical structures while the viral capsid is engulfed by the ER-Golgi intermediate complex (ERGIC) membrane[25]. Although little attention has been given to N's interactions with membranes, these observations suggest that

phosphorylation of N and its membrane interactions are linked and may underlie N's multiple functions. The goal of this paper is to elucidate this link between N phosphorylation and its membrane interaction at the molecular level, providing insights critical for understanding COVID infection.

We sought to understand the phosphorylation-dependent membrane interactions of N through the lens of phase separation. N contains RNA-binding and disordered domains, which enable N and RNA to interact in a dynamic manner. This results in the formation of droplets enriched in protein and RNA, known as biomolecular condensates[3,8,26,27]. N undergoes phase separation with RNA in vitro and in vivo[28,29]; experiments show that phosphorylation of N[19,26], the type of RNA[5,30], and temperature[30] modulate the condensate's propensity to phase separate and affect its molecular dynamics. Biomolecular condensates can display a range of material properties, from liquids to gels, which may be closely associated with their function[31,32]. The material properties of a condensate need not be static, and cellular regulation can modulate material properties and thus condensate

function[33]. Carlson et al. showed that unmodified N and RNA form gel-like condensates and discrete 15-nm particles, while phosphorylated N generates a more liquid-like droplet. They hypothesized that this difference in material properties could be the basis for a dual role of N during the viral lifecycle in both regulating RNA transcription and facilitating nucleocapsid assembly[19,34]. However, how the phosphorylation-dependent material properties of N condensates affect their interaction with membranes has not been studied. Recent investigations on condensate-membrane interactions show that condensate material properties are a key factor determining the mode of interaction between condensates and the membrane surface[35–37]. Based on this, we hypothesized that phosphorylation tunes how N and RNA interact, thus modulating the material properties of N and RNA condensates, consequently influencing condensate-membrane interaction, and in turn allowing N to display the two distinct behaviors observed throughout the viral lifecycle (Fig. 1c, d).

Here, we tested this hypothesis using careful in vitro reconstitutions to map how condensates formed by N and viral RNA fragments interact with membranes. We identified membrane association as a feature of N condensates, and we found that this interaction is regulated at least in part by N's phosphorylation state. Unmodified N condensates retain their shape upon interaction with membranes, but phosphorylation reduces condensate viscoelasticity, such that phosphorylated N (pN) condensates are softer and therefore wet membrane surfaces. Phosphorylation also switches N's ability to interact with viral membrane proteins. These results provide a mechanism by which N can toggle between distinct membrane-associated functions. To understand these changes at a molecular level, we performed structural studies and molecular simulations that revealed that phosphorylation strengthens interactions between pN monomers within pN dimers and thus weakens protein–RNA binding. We propose a model where phosphorylation acts as a functional switch during the viral lifecycle, shifting N's roles from RNA replication to virion assembly by toggling its membrane interaction.

## Results

We reconstituted N or pN and RNA condensates in vitro. We prepared recombinantly expressed N that was either kept unmodified or was phosphorylated in vitro using GSK-3β and SRPK1 kinases[17] (Fig. 1b). Up to nine phosphorylation sites on N were confirmed using mass spectrometry and Phos-Tag SDS-PAGE (Supplementary Fig. 1). We used in vitro transcription to make fragments of viral RNA that are known to promote phase separation of N (Fig. 1e)[30]. First, we tested a fragment containing the first 1000 base pairs from the 5′ end of the viral genome that contains important regulatory information[1]. Second, we tested the fragment of RNA that encodes N (containing the first 75 nucleotides of the 5′ untranslated region recombined onto the N protein coding sequence), given that it is highly produced during viral infection[1]. As expected, mixing N or pN protein with either RNA at 37 °C resulted in their condensation into protein- and RNA-rich droplets (Fig. 1f). A change in morphology with regards to the degree of sphericity of droplets already suggests that condensates are modulated by both type of RNA and phosphorylation status of the protein (Supplementary Fig. 2), two factors that we investigate in depth.

### N condensate composition determines interaction with membranes

Prior work has shown that following viral infection, N accumulates in thin layers around folded ER membranes that are likely vROs[24]. In contrast, later during viral budding events, N is part of RNPs linked by the viral genome[25]. These RNPs do not fuse or grow during the budding process, instead remaining as distinct complexes as the nucleocapsid is engulfed by the ERGIC membrane, forming new

virions[25,38]. We first asked whether we could reproduce these two behaviors of N in vitro.

We developed a system to study the interaction between N and RNA condensates and membranes. We modeled membranes using giant unilamellar vesicles (GUVs). GUVs were made with a lipid composition meant to approximate the human ER membrane[39,40], with 60% DOPC, 25% DOPE, 10% DOPS, and the addition of 5% Ni-NTA lipids. The Ni-NTA lipids allowed us to tether membrane protein fragments to the GUV surface[41] to investigate whether the presence of either of two viral membrane proteins would affect condensate-GUV interaction (Fig. 2a). The viral non-structural protein 3 (Nsp3) localizes to vROs[42] and was suggested to both help form the double layer of membranes as well as form pores that span the membranes[43–45]. Nsp3 is a known interactor of N[46,47] and therefore is a logical candidate for modulating N's interaction with the membranes enveloping vROs. We also studied the viral M protein that is present in the ERGIC membrane, where new virions form[48]. M and N are also known interactors, where M is thought to anchor the RNP complex to the membrane during new virion formation[49,50]. Given the challenges of producing and inserting transmembrane proteins into vesicles, we chose to study the domains of the proteins that are known to interact with N. For M, we used a construct with the C-terminal endodomain of the M protein, fused to a 6xHis tag and GFP[50]. For Nsp3, we took its ubiquitin-like domain 1 (Ubl1)[46], and produced a construct with it fused to a 6xHis tag and GFP.

For this experiment, we used condensates composed of N or pN and the 1–1000 RNA fragment. We added a small amount of GUVs and membrane protein fragments to the samples− either Nsp3 or M fragments or a control solution (Fig. 2b). We observed the GFP signal from the membrane proteins become localized to the surface of GUVs, with a similar degree of tethering of Nsp3 and M fragments to the GUV surface (Supplementary Fig. 3). Next, we used optical tweezers to control the position of condensates, moving them to and holding them at the GUV surface. We then attempted to pull the condensates off the surface and observed whether they remained bound or were mobile. For example, in a sample with pN, 1–1000 RNA, GUVs, and Nsp3 fragments, we brought condensates composed of pN and 1–1000 RNA to the GUV and observed as the condensates wet the surface (Fig. 2c).

We defined five types of interactions between condensates and membranes (Fig. 2d legend): (1) With no interaction, the optical tweezer can move condensates away from a surface it was in contact with. (2) With only binding, we observe the condensates remain attached to the membrane surface even when the optical tweezer is attempting to dissociate the two, but no additional condensate-membrane interactions occur. (3) During membrane wrapping, the condensate binds to the membrane, and the membrane surrounds the condensate over time. (4) In partial wetting, the condensate not only binds to the membrane surface but also partially deforms, expanding the area of contact between the condensate and membrane. (5) Finally, in complete wetting, the condensates totally wet the membrane surface, forming a thin layer of protein and RNA condensate.

We found that if no fragments of membrane proteins (Nsp3 or M) are present, no interaction occurs between N and RNA condensates and the GUV surface, regardless of the phosphorylation status of N (Fig. 2d and Supplementary Movies 1, 2). This suggests that N or pN and RNA condensates have no intrinsic ability to interact with lipid membranes of the composition tested (60% DOPC, 25% DOPE, 10% DOPS, and 5% DGS-NTA(Ni), in 150 mM NaCl, 20 mM Tris-HCl, pH 7.5 buffer). In contrast, when the Nsp3 fragment is at the surface of GUVs, we observe condensates composed of both N and pN binding to the membrane surface (Supplementary Movies 3, 4). With unmodified N, condensates with 1–1000 RNA only partially wet the surface. Condensates with phosphorylated N completely wet the surface, forming a thin layer of condensed material. This behavior of pN and RNA condensates resembles the formation of a layer of protein on vROs that is

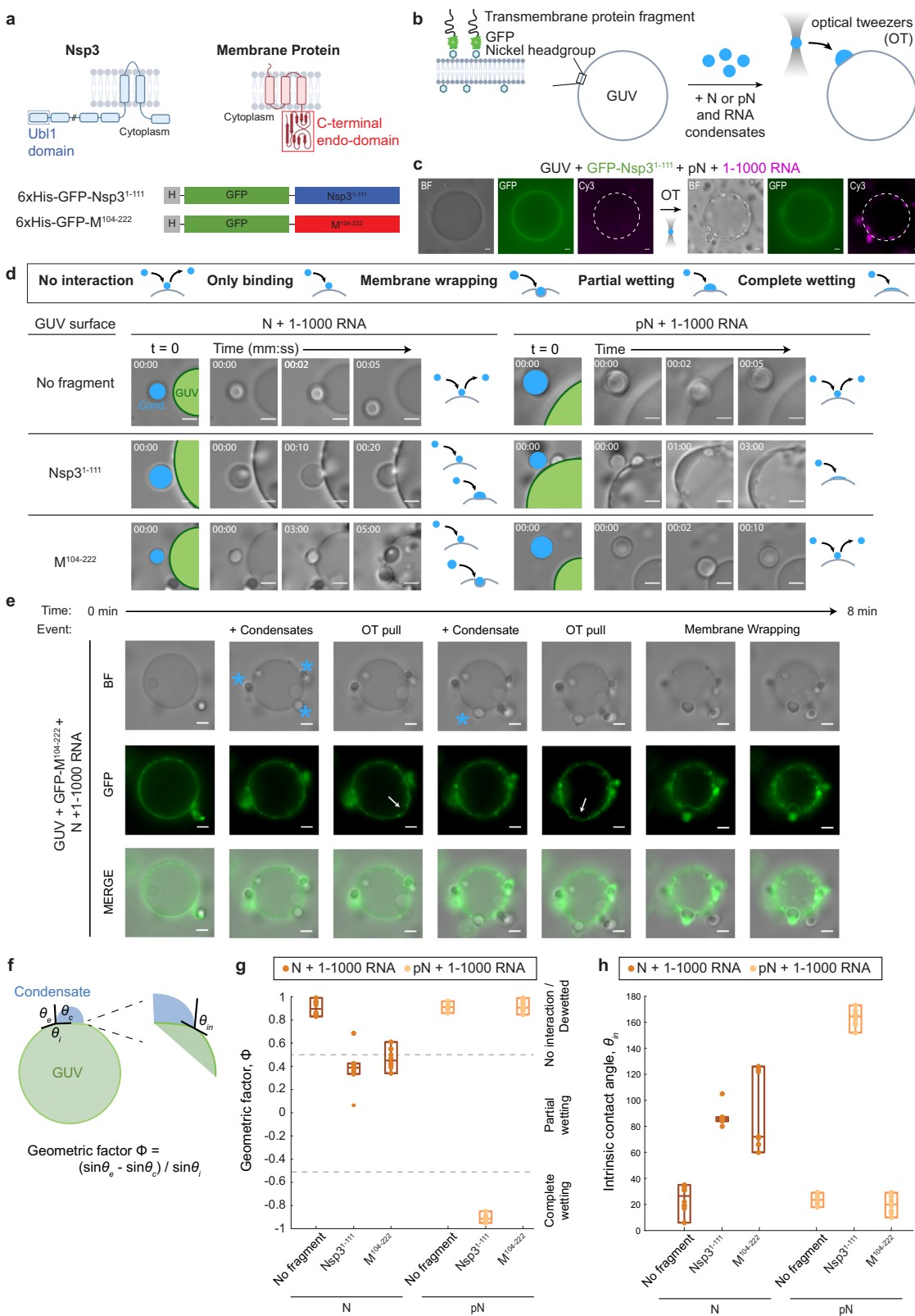

observed in infected cells[24]. Importantly, our results hint that by modulating the material properties of N condensates, phosphorylation shifts the behavior observed from partial to complete wetting. The thin layer of N that is observed surrounding vROs in infected cells may be condensed pN wetting the surface due to pN-Nsp3 interactions.

When we repeated the experiment with the M fragment, we observed an important difference in binding based on the status of N

phosphorylation: while condensates with unmodified N bound to the M-coated GUV surface, condensates with pN did not interact with the surface (Supplementary Movies 5, 6). Therefore, phosphorylation of N may have unexpected effects beyond modulating condensate material properties: it can act as a switch to prevent binding of pN to membrane surfaces displaying the viral Membrane protein, where new virion formation typically occurs. (We investigate how phosphorylation affects

**Fig. 2 | N condensate interaction with membranes. a** Fragments of the membrane proteins used to test membrane interactions: C-terminal endodomain of the Membrane (M) protein and the ubiquitin-like domain (Ubl1) of the Nsp3 protein. Both protein fragments are fused to a 6x Histidine tag and GFP. Schematic created in BioRender. Favetta, B. (2025) https://BioRender.com/s97t568. **b** Schematic of the experimental setup showing the M or Nsp3 protein fragment tethered to the giant unilamellar vesicle (GUV) surface. Condensates composed of 40 μM N or pN plus 300 nM RNA are moved to the GUV surface using optical tweezers. Schematic created in BioRender. Favetta, B. (2025) https://BioRender.com/we3zbf2. **c** Representative widefield images showing that Nsp3 fragments preferentially localize to the GUV surface (left) and condensates composed of pN and 1–1000 RNA wet the surface of GUVs after being delivered to the surface using optical tweezers (OT) (right). White dashed line added to denote the GUV surface. **d** Representative widefield images showing the interaction between condensates and membranes over time, for N vs. pN, and comparing GUVs with no membrane protein, M fragment, and Nsp3 fragment. GUVs are labeled in green, and condensates (Cond.), in blue. Interaction type is qualitatively classified. Schematic created in BioRender. Favetta, B. (2025) https://BioRender.com/654vtk2. **e** Timelapse imaging showing several condensates (blue stars) being dragged by

optical tweezers to the surface of a GUV, to which they bind. Optical tweezers can be used to pull bound condensates which results in deformation of the GUV surface (white arrows). Membrane wrapping occurs in one case. **f** Sketch showing the parameters that define the geometric factor ($\Phi$) and intrinsic contact angle ($\theta_{in}$). Interaction leads to the formation of a contact line between the condensate interface and membrane. Contact angles $\theta_c + \theta_e + \theta_i = 360°$ and we can calculate $\Phi$ to describe the degree of wetting. Wetting can also be characterized by $\theta_{in}$ between condensate and membrane surfaces. Schematic created in BioRender. Favetta, B. (2025) https://BioRender.com/j5wt6ys. **g** Quantification of the geometric factor from each combination of condensate composition and membrane surface. Interactions that lead to membrane wrapping, partial wetting, or complete wetting result in reduced geometric factors. The box plots indicate the median, lower quartile, and upper quartile from $n = 10$ condensate-membrane interactions from $n = 3$ independent trials. Individual data points are shown for each interaction event. **h** Intrinsic contact angle from each combination of condensate composition and membrane surface, calculated from the same image data set as **g**. The box plots indicate the median, lower quartile, and upper quartile. Scale bars = 5 μm. Source data are provided as a Source Data file.

protein–protein interactions between N and M or Nsp3 in Fig. 3). When condensates with N and 1–1000 RNA interacted with M-coated GUVs, we observed binding (6/10 events) or membrane wrapping (4/10 events) (Fig. 2e), the difference between either case likely driven by the membrane tension of the GUVs. In the cases where membrane wrapping occurs, the protein and RNA condensates are partially engulfed into GUVs in the time scale of minutes. The events of membrane wrapping are especially intriguing, given that they resemble the direction of membrane bending required for encapsidation. These observations are consistent with prior EM studies showing that RNPs may contribute to membrane bending after recruitment to the membrane by M[38,51].

Given the important role that cholesterol plays in determining membrane fluidity, and recent evidence pointing to cholesterol's modulation of condensate-membrane interactions[52], we sought to test the effect of adding cholesterol to our GUV membranes. Viral replication organelles originate from ER membranes, which are known to be low in cholesterol content (~2–5%)[40]. Although the SARS-CoV-2 virus buds from the nearby ERGIC[53], their membranes were found to include much higher percentages of cholesterol (~25%)[54]. Therefore, we tested the effect of inclusion of 2, 5, and 25% cholesterol on GUV membranes in place of DOPC (Supplementary Fig. 4). Addition of cholesterol did not affect the binding interaction of condensates with membranes lacking viral membrane fragments. Next, we observed that the addition of cholesterol reduced the degree of condensate wetting in cases where the GUV was coated with Nsp3 fragments, in a concentration-dependent manner. However, as noted above, the ER-derived vRO membrane where N-Nsp3 interaction occurs is known to have low cholesterol content, and therefore, we speculate that cholesterol has a minor effect on N wetting to viral replication organelles. Finally, inclusion of cholesterol did not affect the type of condensate-membrane interaction observed with the M fragment present. We conclude that although cholesterol can affect condensate-membrane interaction, in the particular cases that are most physiologically relevant to SARS-CoV-2 infection, addition of cholesterol did not change the interaction type observed.

We quantified the affinity between condensates and GUVs by measuring the geometric factor that considers the contact angles along the contact line between the GUV surface, the condensate, and the external solution, as well as the intrinsic contact angle between condensate and GUV (Fig. 2f–h)[37]. These are determined by the material properties of the condensate, the strength of interaction between the condensate and the membrane surface, and the membrane tension of the GUV. No interaction between condensates and the membrane results in the highest geometric factors. An intermediate geometric factor is found with N condensates and Nsp3 fragments,

where partial wetting is observed, and N condensates with M fragments, where either binding or membrane wrapping is observed. A drastic reduction in geometric factor occurs with pN and Nsp3, where complete wetting occurs. The different degrees of interaction also impact the intrinsic contact angle between condensate and GUV. Partial wetting or membrane wrapping causes an intermediate reduction in the angle, while complete wetting causes a drastic reduction in the angle (Fig. 2h). Overall, with this panel of condensate compositions and membrane surfaces, we were able to reproduce several behaviors that N displays during the viral lifecycle.

## Phosphorylated N cannot bind to the SARS-CoV-2 Membrane protein

Given our observation that pN condensates did not bind to GUVs with M fragments, we examined whether phosphorylation affects protein–protein interactions between N and M or Nsp3. Several groups have sought to understand the binding mechanism between coronavirus N and M proteins[49,50], but no information is available on the effect of phosphorylation of N on its interaction with M. A stretch of amino acids (168–208) within the linker region of N was identified as critical for N–M interactions in the SARS-CoV-1 virus[49]. More recently, the linker domain of the SARS-CoV-2 N was shown to be necessary for N to co-phase separate with M[50]. Together, these studies point to a potential role of the SR-rich domain of N in interacting with M. Our results from Fig. 2 suggest that phosphorylation of the SR region of N may inhibit binding between N and M.

We performed a partitioning experiment using confocal microscopy (Fig. 3a). We added the fluorescently tagged M fragment or Nsp3 fragment to N or pN and observed the partitioning of the membrane protein fragments into condensates (Fig. 3b, c). We confirmed that M has a much stronger affinity to unmodified N than to pN. Unmodified N binds to M independent of the presence of viral RNA fragments, though phase separation is promoted by the presence of 1–1000 RNA. We observed contrasting results with phosphorylated N mixed with M. First, M does not drive the condensation of pN without the addition of RNA. Second, M does not partition preferentially into condensates composed of pN and viral RNA (partition coefficient for M is $9.8 \pm 0.3$ in N + 1–1000 RNA condensates vs. $0.3 \pm 0.1$ in pN + 1–1000 RNA condensates; partition coefficient is defined as the ratio of average fluorescence intensity inside vs. outside the condensates). Given that M likely interacts with residues around N's SR domain[49], it is not surprising that phosphorylation of the SR region affects binding between the two proteins. These results are intriguing because they suggest a mechanism for the timing of viral assembly. As noted above, a majority

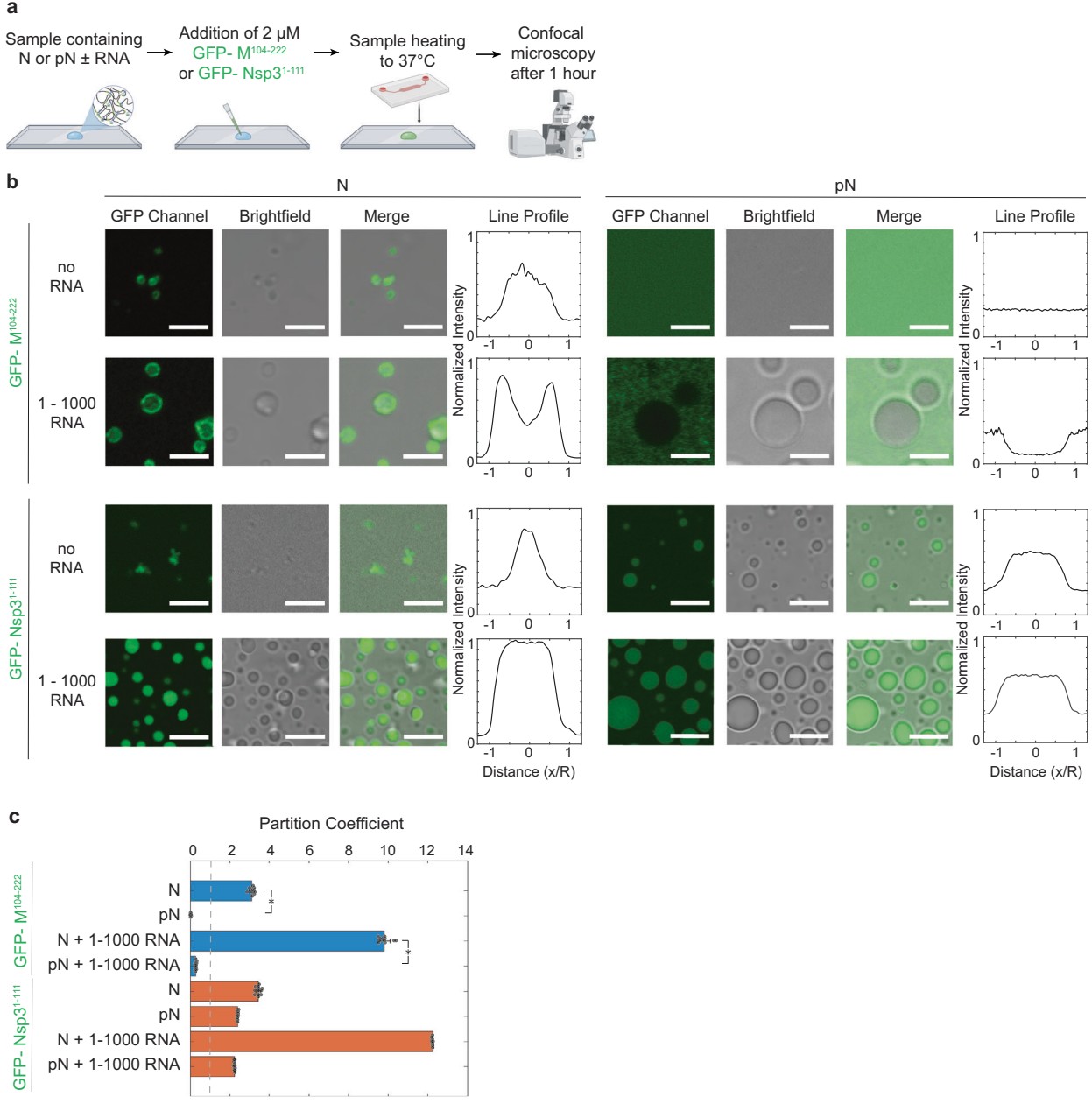

**Fig. 3 | N and pN interaction with membrane proteins, M and Nsp3. a** Schematic of the experimental setup where 6xHis-GFP-M$^{104-222}$ or 6xHis-GFP-Nsp3$^{1-111}$ is added to a condensate sample, and partitioning of GFP-tagged protein into condensates is quantified using confocal microscopy. Schematic created in BioRender. Favetta, B. (2025) https://BioRender.com/o28w195. **b** Representative confocal images from condensates composed of 40 μM N or pN plus either no RNA or 300 nM RNA; plus either 2 μM 6xHis-GFP-M$^{104-222}$ or 6xHis-GFP-Nsp3$^{1-111}$. 6xHis-GFP-M$^{104-222}$ binds to N but not pN condensates, while 6xHis-GFP-Nsp3$^{1-111}$ partitions in regardless of the choice of N vs. pN. Normalized line profiles across condensates,

representing averages of at least $n = 20$ condensates from three independent trials. Scale bars = 5 μm. **c** Quantification of the partitioning of GFP-tagged proteins into N or pN condensates by dividing the average fluorescence inside condensates by the average background fluorescence. Because condensates did not form in samples with pN plus 6xHis-GFP-M$^{104-222}$ with no RNA, the partition coefficient was defined as 0. Data presented as mean values ± SD, $n = 10$ images from $n = 3$ independent trials. $p$ values were determined using two-way ANOVA followed by post hoc Tukey's test. *$p < 0.0001$. Source data are provided as a Source Data file.

of N within infected cells is phosphorylated, but N included within new virions is unmodified[16]. pN may not be incorporated into newly formed virions because this protein is unable to bind M and anchor the RNP to the membrane at the site of viral assembly.

As a comparison, we also investigated whether phosphorylation affects the partitioning of the Nsp3 protein's Ubl1 domain into N or pN condensates. Previous reports showed that the Nsp3 Ubl1 domain partitions into condensates composed of both N and a phosphomimetic version of N[19]. Our results qualitatively agree

with these findings, although we found pN has a lower affinity than N for Nsp3, based on our measured partition coefficients (12.6 ± 0.1 for Nsp3 in N + 1–1000 RNA condensates vs. 2.2 ± 0.1 in pN + 1–1000 RNA condensates). Given the hypothesized role of N in delivering the viral genome to Nsp3-coated replication organelles following viral entry into cells, and given that the films of N observed adhering to replication organelles[24] are likely composed of phosphorylated N, it is not surprising that both unmodified and phosphorylated N bind to Nsp3.

## Phosphorylation of N modulates the material properties of N condensates

Based on previous data on the liquid-like properties of pN condensates[19] and our membrane binding experiments, we hypothesized that the dynamic nature of pN condensates is what enables them to relax at the surface of GUVs. To explore the material properties of condensates, we first used fluorescence recovery after photobleaching (FRAP). We photobleached either fluorescently labeled protein or RNA within a region of the condensate and observed how the fluorescence recovered over time. The advantage of using FRAP here is that it allows us to independently assess the dynamics of protein and RNA in our multi-component condensates. N bound to 1–1000 RNA recovers more slowly than pN bound to the same RNA (recovery half-life $\tau = 2.6 \pm 0.1$ min for N and $1.7 \pm 0.1$ min for pN) (Fig. 4a, b and Supplementary Fig. 5). This is in agreement with previous data showing a more rapid FRAP recovery in condensates with a phosphomimetic version of N compared to unphosphorylated N[19]. In addition, we also observe that the total recovery of N is lower compared to pN (62 vs. 94%), suggesting a pool of N is fixed to the RNA. Both metrics suggest that pN has a greater mobility than N when bound to 1–1000 RNA. In contrast to the results obtained by photobleaching protein within condensates, we observed almost no recovery of the fluorescence when RNA was bleached (Fig. 4a, c), independent of protein phosphorylation status. RNA appears to form a network onto which protein can bind and dissociate[55]. Individual RNA molecules may move locally but do not appear to diffuse long distances, likely due to their ability to form intermolecular base pairs. Overall, our FRAP results support our hypothesis that differences in material properties contribute in part to the type of condensate-membrane interaction observed. For example, condensates composed of pN (which can wet surfaces) showed a greater protein recovery in FRAP experiments, suggesting that the dynamic movement of pN allows the condensates to reorganize at the surface of membranes over time.

To quantitatively assess how phosphorylation affects N condensate material properties, we turned to passive microrheology. In passive microrheology, we embed 500 nm fluorescent tracer beads into condensates and track their movement. The mean squared displacement (MSD) of beads depends on their viscous and elastic environment[56]. Beads embedded within condensates composed of pN + 1–1000 RNA displayed a greater MSD at all lag times when compared to beads embedded in N + 1–1000 RNA condensates (Fig. 4d and Supplementary Fig. 6). The MSD of beads did not increase linearly with lag time for all samples, revealing that some condensates behave as viscoelastic fluids under the experimental conditions. Using the Generalized Stokes-Einstein Relation, we estimated viscous and elastic moduli. For N + 1–1000 RNA condensates, the elastic modulus dominates the viscous modulus at high frequencies (>4 Hz). In contrast, for pN + 1–1000 RNA, the viscous modulus is dominant for all frequencies measured (Fig. 4e, f). We then quantified the zero-shear viscosity of the samples, i.e., the limiting value for viscosity at low frequencies. For condensates composed of N + 1–1000 RNA, we found a viscosity of $192 \pm 3.6$ Pa·s, while for condensates with pN, we measured a viscosity of $59 \pm 3.4$ Pa·s, representing a ~3x reduction in viscosity (Fig. 4g and Supplementary Fig. 7).

The change in viscosity upon phosphorylation of N may be due to modulation of protein–protein and/or protein–RNA interactions, the latter via either specific or nonspecific binding to RNA. We assessed each of these possibilities by testing the material properties of N or pN condensates with unstructured RNA (polyrA) or with 5% PEG of average molecular weight 8000 Da, which acts as a crowding agent to drive N phase separation without RNA (Supplementary Fig. 2). Phosphorylation of N decreased the viscosity of condensates with polyrA from $64 \pm 4.2$ Pa·s to $12 \pm 0.4$ Pa·s, suggesting that nonspecific interactions between protein and RNA were disrupted. Furthermore, phosphorylation decreased viscosity of condensates with PEG from $11 \pm 0.7$ Pa·s to $4 \pm 0.3$ Pa·s, suggesting that protein–protein interactions are also disrupted to some degree (Fig. 4g).

We also quantified the terminal relaxation time of the condensate network, which is the inverse of the crossover frequency between the predominantly elastic and predominantly viscous regimes. At timescales below the relaxation time, the condensates behave as elastic materials. Elasticity is reduced by phosphorylation in both polyrA and 1–1000 RNA samples (Fig. 4h). Condensates with N and polyrA or 1–1000 RNA have a relaxation time of 0.02 and 0.3 s, respectively, while condensates with pN have no measurable relaxation timescale, suggesting that phosphorylation disrupts the ability of N and RNA to crosslink (Supplementary Fig. 8). The elasticity displayed by complexes formed between unmodified N and RNA may have a biological function in protecting the RNA from mechanical stress[57].

We confirmed our material property measurements using micropipette aspiration (MPA)[58,59]. We recorded how the length of the aspirated condensate in a micropipette changes as a function of time in response to an applied pressure. We quantified the viscosity for the different protein combinations and obtained the same trend and rank order that was observed using microrheology (Fig. 4i and Supplementary Fig. 9). Viscosities quantified via MPA were consistently lower than the zero-shear viscosities obtained from microrheology; when the two data sets are plotted against each other, they have a linear fit with slope of 0.73 and $R^2$ of 0.99. Notably, MPA measurements also showed that pN condensates have lower viscosity than the corresponding N condensates. Together, our microrheology and micropipette aspiration results point to N and RNA condensates being viscoelastic fluids whose properties depend on both RNA type and phosphorylation status. Phosphorylation of N appears to loosen the network of interactions between N and RNA, thus reducing the viscosity and elasticity of condensates.

## RNA type also modulates condensate material properties

We repeated experiments with a second viral RNA fragment to assess whether our results depend on RNA sequence and structure. RNA sequence and structure are known to affect the material properties of other condensates, though most quantitative studies so far have focused on shorter, artificial RNA sequences[60,61]. For N protein, evidence exists that the morphology of condensates depends on the type of RNA included[30], however, the effect on condensate material properties has not been investigated. Here, we compare our results from 1–1000 RNA with N RNA, introduced previously, which encodes the N protein and is 1340 base pairs long. N RNA is an important viral RNA to test because it is abundant in infected cells, and from a biophysical perspective, it is (1) longer than 1–1000 RNA, and as such has a higher ensemble diversity (which describes the diversity of conformations the RNA is predicted to fold into; it is predicted to be 198.5 for N RNA vs. 148.1 for 1–1000 RNA, based on ViennaRNA[62]) and (2) contains a different pattern of preferred binding sites for N[30]. These factors suggest that N RNA forms a more entangled network in condensates compared to 1–1000 RNA, resulting in the different condensate morphologies observed (Fig. 1f). We found similar behavior between condensates with 1–1000 or N RNA (Supplementary Fig. 10), both in terms of binding to GUVs with different membrane protein fragments, as well as their material response to phosphorylation (phosphorylation reduces viscosity and elasticity). These results suggest that the change in N condensate material properties and membrane interaction upon phosphorylation are not RNA structure dependent. Across experiments, we did observe that condensates with the longer N RNA were less likely to deform than their counterparts with 1–1000 RNA. This data hints that complexes of unmodified N and longer fragments of RNA, such as genomic RNA, would appear solid-like and with an important elastic response, which may play a role in mechanically protecting RNA within virions[57]. To further study these questions, additional analyses with varying RNA lengths and structures would be required.

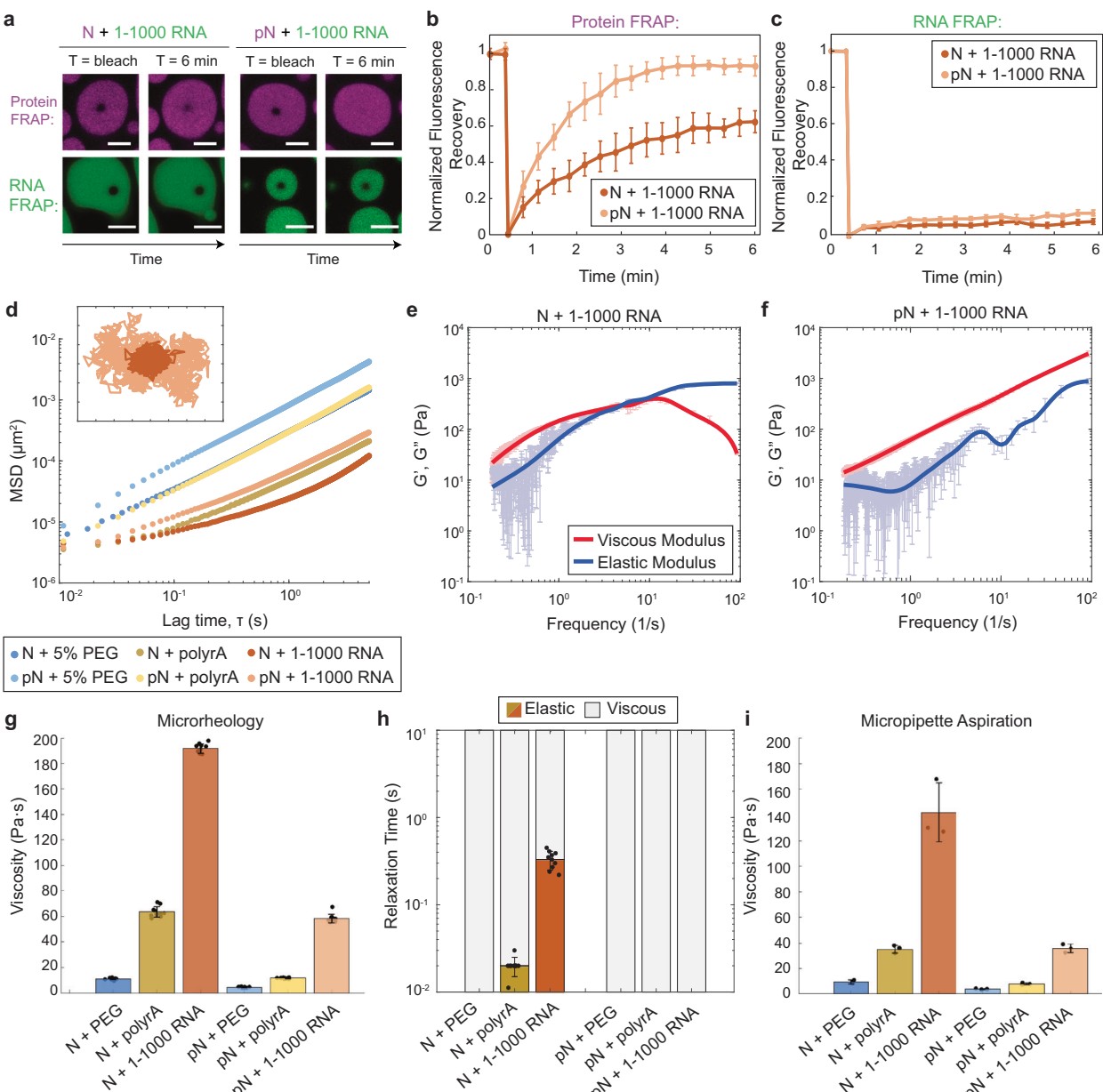

**Fig. 4 | Phosphorylation modulates material properties of N and RNA condensates. a** Fluorescence recovery after photobleaching (FRAP) of N and RNA condensates, in which 5% of N/pN is labeled with Alexa-647 and 5% of RNA is labeled with Cy3. pN recovers to a greater extent over time when compared to N, while in neither case does RNA signal recover. Scale bar = 5 μm. **b** Quantification of the FRAP of protein in N vs. pN condensates. Data presented as mean values ± SD, $n = 3$ independent trials. **c** Quantification of the FRAP of RNA in N vs. pN condensates. Data presented as mean values ± SD, $n = 3$ independent trials. **d** Ensemble MSD versus lag time (prior to noise correction) for the protein and RNA combinations tested in this study. Inset: Representative trajectories from two-dimensional particle tracking showing Brownian motion of beads in N vs. pN condensates with 1–1000 RNA. Each tick represents 5 nm. **e** Plot with the viscous modulus (G", red) and elastic modulus (G', blue) of N + 1–1000 RNA condensates as calculated from the MSD after noise correction. Data presented as mean values ± SD, $n = 10$ different videos from $n = 3$ independent trials. **f** The viscous and elastic moduli after noise correction for pN + 1–1000 RNA condensates showing no crossover frequency in the range studied. Data presented as mean values ± SD, $n = 10$ different videos from $n = 3$ independent trials. **g** The zero-shear viscosity of the protein and RNA condensates studied, calculated from the particle-tracking results after noise correction. Data presented as mean values ± SD, $n = 10$ different videos from $n = 3$ independent trials. **h** Quantification of the timescales at which the elastic modulus dominates (color) versus the viscous modulus dominates (gray) in protein and RNA condensates. Data presented as mean values ± SD, $n = 10$ different videos from $n = 3$ independent trials. **i** Viscosity of the protein and RNA condensates from micropipette aspiration. Data presented as mean values ± SD, $n = 3$ independent trials. Source data are provided as a Source Data file.

## Phosphorylation weakens binding between N and RNA by promoting intra-dimer interactions

We have shown that phosphorylation of N results in condensates that more readily deform at membrane surfaces and that have lower viscosity and elasticity, both likely driven by a loosening of the interaction network between protein and RNA. This change in interaction does not

appear to be dependent on specific RNA structures, as phosphorylation also drastically softens condensates with unstructured polyrA (Fig. 4g–i). We therefore asked how phosphorylation affects N, such that the interaction between protein and RNA is diminished.

We leveraged fluorescence polarization to measure the binding affinity between the N protein and fluorescently labeled RNA (Fig. 5a,

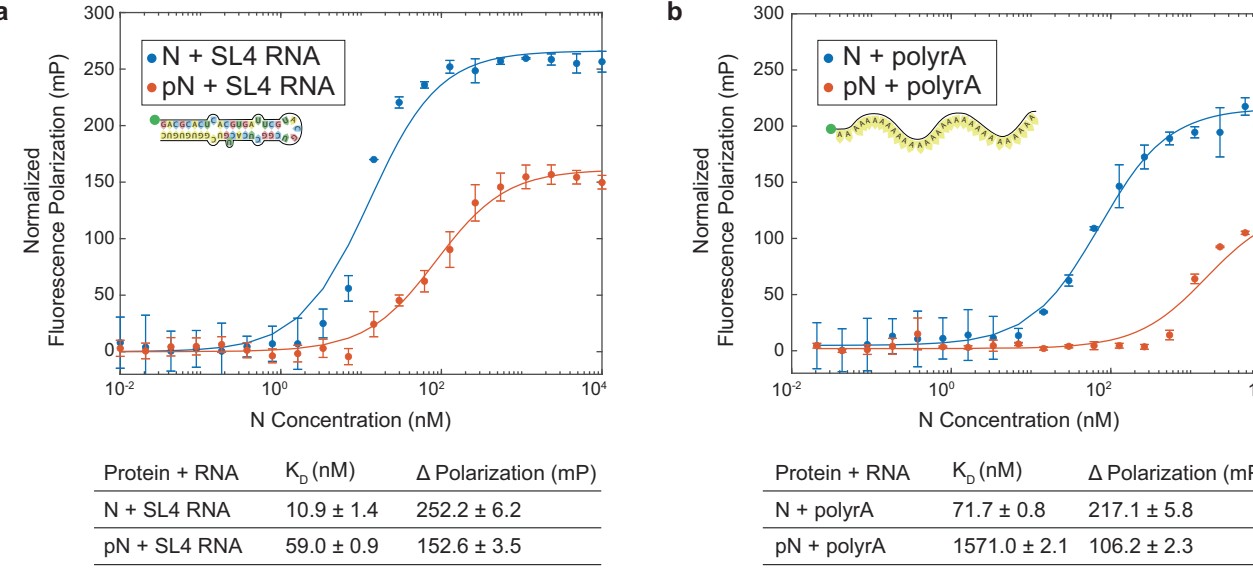

**Fig. 5 | N protein phosphorylation weakens RNA binding affinity.**
**a** Quantification of binding affinity between N vs. pN and the viral stem loop 4 RNA (SL4) based on a change in normalized fluorescence polarization (minimum polarization set to 0). Data presented as mean values ± SD, *n* = 3 independent

trials. **b** Quantification of binding affinity between N vs. pN and unstructured 30-base polyrA from normalized fluorescence polarization. Data presented as mean values ± SD, *n* = 3 independent trials. Source data are provided as a Source Data file.

b). First, we assessed whether phosphorylation affects the binding of N to a stem-loop structure (SL4) present in the 1–1000 RNA fragment. SL4 was identified as a preferred binding site for N's RNA binding domain[5,30]. Phosphorylation of N reduces the binding affinity to SL4 RNA, increasing the dissociation constant ($K_D$) from $10.9 \pm 1.4$ nM to $59.0 \pm 0.9$ nM. Although this is a six-fold reduction in binding affinity, the concentration of N at which our membrane interaction and material property experiments were conducted are well above the nanomolar $K_D$s measured, and therefore, this change in binding cannot explain the material property changes observed. Next, to test the hypothesis that nonspecific binding between protein and RNA is disrupted by phosphorylation, we measured the binding affinity between N or pN and a 30-base polyrA. Phosphorylation causes a 20-fold change in $K_D$ from $71.7 \pm 0.8$ to $1571.0 \pm 2.1$ nM, larger than that observed for the SL4 RNA. We conclude that phosphorylation reduces N's interaction with unstructured RNA more so than structured RNA. Importantly, for both polyrA and SL4, phosphorylation of N also reduced the maximum polarization. This suggests a greater rotational freedom in the protein–RNA complexes formed with pN that could be driven either by the formation of a more compact complex or a conformation in which the protein-bound RNA retains more rotational mobility[63]. We anticipated that phosphorylation would weaken binding between N and RNA due to electrostatic repulsion. However, our fluorescence polarization results suggest that phosphorylation may also be inducing a conformational change in N that affects its RNA-binding properties.

To assess the conformational change of the N protein dimer following phosphorylation, we used small-angle X-ray scattering (SAXS), a technique compatible with the size and largely disordered nature of N protein. Our SAXS results indeed show that N and pN are partially disordered dimers in solution. This is in agreement with previously published data on unmodified N protein[9]; SAXS of pN had not been previously reported. With phosphorylation, we found a significant decrease in radius of gyration ($R_g$), from $60.2 \pm 1.3$ Å for N to $54.4 \pm 2.0$ Å for pN, a finding supported by both the pair distance distribution and the Guinier approximation (the decrease in $R_g$ was consistent across concentrations) (Fig. 6a, b and Supplementary Figs. 11, 12). The shape of the pair distance distribution function also changes with phosphorylation. The presence of multiple peaks in the

unmodified N data highlights the multidomain nature of the N protein dimer[64]. The lack of these peaks in the phosphorylated N data suggests the domains can no longer be resolved, indicating there may be new interactions bringing the different parts of the protein together. Next, analyzing normalized Kratky plots confirms both N and pN display behavior typical of a multidomain protein with flexible linkers (Fig. 6c)[65]. The elevated tail for N indicates greater extension of the protein compared to pN. Using the scattering data to construct bead models for N and pN allows us to visualize these changes (Fig. 6d, e and Supplementary Fig. 13). Our bead models show a new region of electron density between monomers within pN dimers, suggesting that indeed new interactions within pN dimers take place that are not present in N dimers.

To better understand how the conformation of N changes upon phosphorylation, we used single-molecule Förster resonance energy transfer (smFRET). First, for full-length N, we confirmed that phosphorylation does not affect the conformation of the dimerization domain[66] or the tendency of N to dimerize (Supplementary Fig. 14). Next, we used smFRET to study the conformation of the linker region. We probed two constructs of N with fluorescent labels flanking the linker region (at residues 172 and 245): full-length N and truncated N[1-246], which lacks the dimerization and C-terminal disordered domains. We performed experiments at two protein concentrations: low concentration (100 pM labeled protein), at which N is in its monomeric form, and high concentration (100 pM labeled protein + 1 µM unlabeled protein for full-length N or 4 µM unlabeled protein for N[1-246]), at which dimers form if the dimerization domain is present. We measured the distribution of transfer efficiencies for each protein construct at each concentration (Supplementary Fig. 15). The distributions represent a dynamic ensemble of conformations, as supported by the corresponding analysis of donor lifetime vs. transfer efficiency (Supplementary Fig. 15). Therefore, we used the mean transfer efficiencies to calculate the root mean square distance (RMSD) between labeled residues, which can be compared across samples to understand the degree of expansion of the linker (Supplementary Fig. 15). Consistent with previous measurements of the same constructs[8], we observed a mean transfer efficiency across the linker of ~0.6 for full-length N and

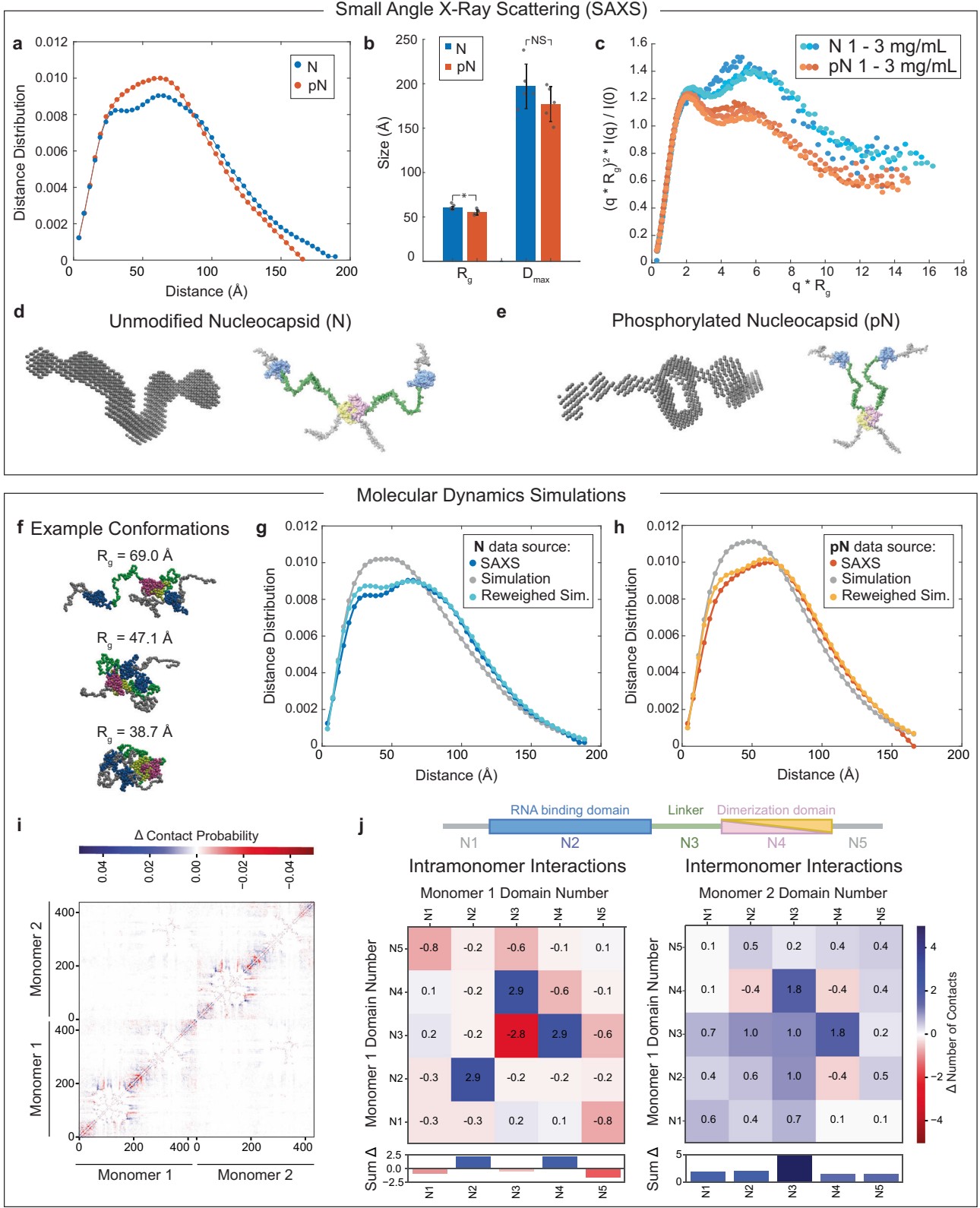

0.75 for $N^{1-246}$. We found that phosphorylation causes an expansion of the linker region, which was unexpected given that we previously measured by SAXS a reduction in radius of gyration of the N dimer with phosphorylation. However, analysis of the sequence charge decoration (SCD) of the linker supports our smFRET findings and suggests that addition of several negative charges to the domain causes local repulsion that results in

expansion[67] (Supplementary Fig. 15). Importantly, our smFRET results show that the degree of expansion due to phosphorylation is greatly reduced in the full-length construct under dimer conditions. To reconcile the SAXS and smFRET results, one possibility is that phosphorylation of the N dimer drives the formation of new interactions across linker domains of the dimer that counteract the expansion caused by phosphorylation.

**Fig. 6 | N protein phosphorylation weakens RNA binding affinity due to change in protein conformation. a** Representative pairwise interatomic distance distribution P(r) derived from SAXS for N and pN. N or pN concentration = 3 mg/mL. **b** The maximum dimension ($D_{max}$) and radius of gyration ($R_g$) for N and pN derived from the pair distance distributions. Data were presented as mean values ± SD, $n = 4$ trials with concentration ranging from 0.5 to 3 mg/mL. $p$ values were determined using two-sided student's $t$-test; asterisk indicates $p = 0.02$. **c** Normalized Kratky plot comparing the scattering of N vs. pN, indicating a structural change has occurred due to phosphorylation. Concentrations shown for N or pN are 1, 1.5, and 3 mg/mL from lighter to darker. **d** Bead model representation for the N dimer developed from SAXS results (left) and hypothesized conformation of N (right). **e** Bead model representation for the pN dimer developed from SAXS results (left) and hypothesized conformation of pN highlighting new intermolecular interactions (right). **f** Snapshots from N simulation depicting conformations that demonstrate the protein's representative collapsed and extended states. The radius of gyration of each selected conformation is noted. **g** Comparison of the pairwise interatomic distance distribution P(r) derived from SAXS and molecular dynamics simulation before and after reweighting with SAXS data for N. **h** Comparison of the pairwise interatomic distance distribution P(r) derived from SAXS and molecular dynamics simulation before and after reweighting with SAXS data for pN. **i** Change in contact probability between all residues within a dimer, compared between N and phosphomimetic N simulations. Lower left and upper right quadrants represent intramonomer interactions, while the upper left and lower right quadrants represent intermonomer interactions. Blue coloring indicates that more frequent interactions occurred in phosphomimetic N compared to unmodified N. **j** Analysis of the difference in intramonomer (left) and inter-monomer (right) interactions between N and phosphomimetic N, per domain of the protein (as defined by the schematic of N above). Positive numbers/blue coloring indicate that more frequent interactions occurred in phosphomimetic N compared to unmodified. Below is the sum of change in interactions across each domain for both intra- and inter-domain interactions. Source data are provided as a Source Data file.

To explore the new interactions that arise upon N protein phosphorylation, we conducted coarse-grained molecular dynamics simulations of the protein. We combined the HPS-Urry model for simulation of disordered proteins[68,69] with an elastic network model and center of mass (COM) bead placement scheme for modeling folded domains[70]. To mimic the effects of phosphorylation, we used phosphomimetic mutations (S to D and T to E). We found reasonable agreement between the radii of gyration measured in SAXS and those predicted in the simulations, which were 55.5 Å for N and 52.4 Å for phosphomimetic N (SAXS results were 60.2 ± 1.3 Å for N to 54.4 ± 2.0 Å for pN). Figure 6f depicts three conformations of the N protein that were captured by the simulation, with a range of radii of gyration. Next, we compared the pair distance distributions from our simulations of N and phosphomimetic N protein with SAXS data (Fig. 6g, h). Although the simulation captured the overall collapse of N upon phosphorylation, it did not fully capture the change in conformation of the protein that we observed through SAXS, which is reflected in the shape of the pair distance distribution function[64]. This indicates that while the simulation model may capture accurate conformations, it lacks the level of detail needed to consistently match experimental data across the full ensemble.

To refine our simulation, we incorporated information from our SAXS experiments. We used a Bayesian Maximum Entropy ensemble reweighting scheme to adjust the statistical weights of the simulated conformations so that their pair distance distribution better matches the experimental SAXS data[71] (Supplementary Fig. 16). This helps identify subsets of conformations that explain the experimental data, which may have been undersampled due to shortcomings in the simulation or missing environmental effects. For example, the phosphomimetic mutations used in simulations lack the additional steric bulk of phosphate groups, and simulations may not fully capture how the ionic conditions in experiments modulate electrostatic interactions involving the phosphate groups[72]. Upon reweighting, the radii of gyration obtained for N and phosphomimetic N more closely matched the values obtained from SAXS, with 58.5 Å for N and 55.0 Å for phosphomimetic N (Supplementary Fig. 17A, B and Supplementary Table 1). The pair distance distribution function from the reweighted simulation also more closely resembled the data obtained from SAXS (Fig. 6f, g). For example, with reweighting, the discrepancy between the pair distance distribution for N reduces from a $\chi^2$ of 7.5 to 2.2. Finally, with reweighting, we also observe agreement between simulations and smFRET results regarding the expansion of the linker domain (Supplementary Table 2).

We then investigated the change in intra- and inter-domain interactions predicted by the simulation with phosphomimetic mutations (Fig. 6i, j). With the simulation, we can quantify the change in the number of contacts between residues both within each monomer and across monomers in a dimer. We found that most intramonomer interactions decrease in the phosphomimetic protein compared to unmodified N, with the exception of N2–N2 and N3–N4 interactions. While the N2 domain was almost entirely held folded by the harmonic constraints of the simulation, the last 13 residues of the domain were left flexible due to their high B-factor in the reference crystal structure. Following the phosphomimetic mutations to the nearby linker (N3), these 13 residues interact with the folded part of the N2 domain at a higher frequency (Supplementary Fig. 17A). Separately, the increase in N3–N4 interactions observed is likely driven by the negative charges added to the N3 domain interacting with a positive patch of residues in the beginning of the N4 folded domain (Supplementary Fig. 17B). Interestingly, we observe a significant decrease in intramonomer interaction within the N3 domain, which matches our smFRET observations. The main driver of this loss of interaction is a decrease in contact between nearby residues of the SR-rich region following the addition of several negative charges due to the phosphomimetic mutations (Supplementary Fig. 17C).

In terms of intermonomer interactions, we found a gain in interaction for almost all combinations of domains of N. In particular, we observe that the linker (N3) gains contacts with the other monomer's N2, N3, and N4 domains. The N3 domain gains interactions with an arginine-rich patch on the N2 domain, and it interacts directly with the SR-rich region of the opposite N3 domain, which is likely also driven by interactions with arginine residues (Supplementary Fig. 17D, E). Finally, the N3 domains interact with a positively charged patch in the N4 domain, which is rich in both arginine and lysine residues (Supplementary Fig. 17F). Therefore, binding between the negatively charged phosphomimetic residues and positively charged arginine and lysine residues are the main source of new interactions within the phosphomimetic N dimers.

Together, our SAXS, smFRET, and simulation results support that a conformational change in N occurs as a result of phosphorylation. Previous molecular simulation of a fragment of N containing the linker domain (N3) had found a phosphorylation-induced increase in intra- and intermolecular contacts due to the formation of salt bridges between phosphate groups and arginine side chains[26]. Our results partly agree with these findings: while intramonomer N3 interactions decrease, binding between arginine residues and phosphate groups across members of a pN dimer appears to be the basis for the new intermonomer interactions observed. Overall, we speculate that new interactions within and between monomers occupy the linker and thus interfere with pN's ability to bind to RNA. The more compact conformation of pN dimers and reduced affinity between protein and RNA[10] explain our earlier observations from the binding affinity assay and material properties measurements. Therefore, this new model ties

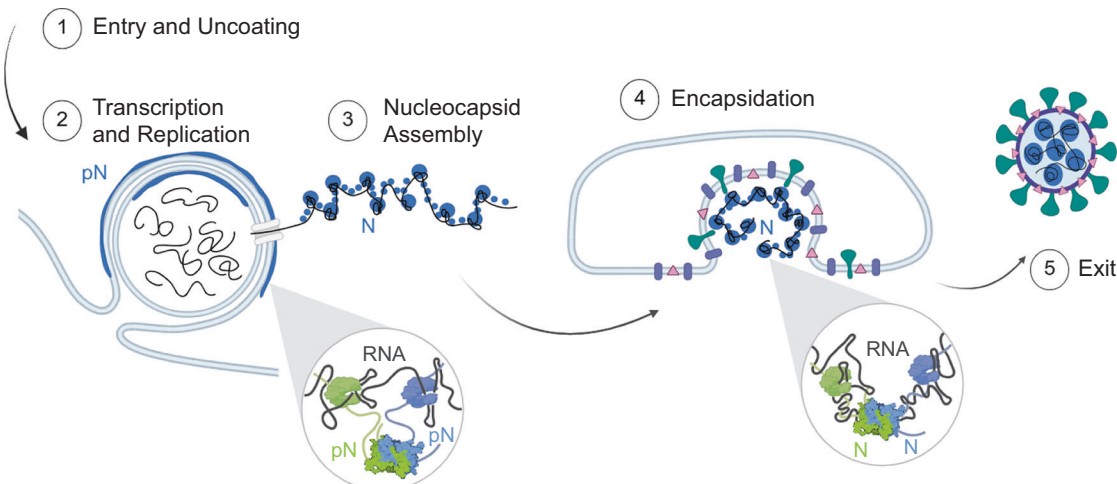

**Fig. 7 | Model of N protein form and membrane-associated state during the viral lifecycle.** Inside the cell (1), the vast majority of N is found in its phosphorylated form, localized to the surface of viral replication organelles (2). Phosphorylation promotes linker–linker interactions across N dimers that weaken the interaction between N and RNA (inset, left). This looser protein–RNA interaction network within condensates results in a relatively low viscosity and elasticity that may facilitate the molecular diffusion needed during RNA transcription and replication. A second population of N that is destined to form new virions binds to new viral genomic RNA (gRNA) and condenses RNA into small spherical complexes (3). This unmodified protein binds tightly to gRNA through both specific and nonspecific interactions (inset, right). These assemblies have high viscosity and elasticity that may support a protective function of N towards gRNA. The capsid, comprised of gRNA and N, is engulfed by the ERGIC membrane, facilitated by N and M interactions (4). New virions exit the infected cells (5). Schematic created in BioRender. Favetta, B. (2025) https://BioRender.com/382j936.

together the conformational change of N upon phosphorylation with its reduced ability to interact with RNA, which in turn affects its condensate material properties and its form when interacting with membranes.

## Discussion

Despite significant interest, there is still limited knowledge about how the SARS-CoV-2 N protein is regulated during viral replication. Our work was motivated by three open questions. First, N is thought to have several functions during viral replication, including regulating viral RNA transcription and translation, and facilitating encapsulation of genomic RNA[2,3]. How can one protein perform such distinct functions? Second, N's regulatory functions likely occur near viral replication organelles[24], while its role in RNA encapsulation is known to take place at the surface of the ERGIC membrane, where viral budding occurs[25]. How does N associate with these different membranes? Third, N is known to phase separate in vitro and when expressed in cells[3,30,50]. However, is phase separation important for the function of N during viral replication? Building upon prior research, we reasoned that phosphorylation holds a clue to answering these three questions. Phosphorylation was previously hypothesized to be the key mechanism of regulation of N during viral replication[19]. Two populations of N exist inside infected cells, with the majority of N being phosphorylated, but with N that is incorporated into new virions remaining unmodified[13–17]. Phosphorylation was also shown to alter the behavior of N condensates[19]. We combined these observations to formulate a new hypothesis: that phosphorylation may be responsible for toggling N between its functions by modulating the material properties of N condensates and thus switching between its two membrane-associated states.

In this work, we reveal membrane association as a key feature of N phase separation. We argue that the phosphorylation state of N determines its two membrane-associated behaviors in infected cells: (1) being adhered to membrane surfaces and (2) being tightly bound to RNA when engulfed into new virions. Our experiments show that phosphorylation modulates the material properties of N and RNA condensates, and thus the type of interaction between condensates and membranes, which is important for the function of N. We next studied N at a molecular level and identified that phosphorylation results in new interactions within and between the monomers of a dimer, thus interfering with the protein's ability to interact with RNA. We propose a model that ties together an understanding of how phosphorylation acts as a switch controlling the conformation of N and its behavior during viral replication (Fig. 7). In its phosphorylated form, N prefers intermolecular interactions between N monomers within a dimer over interaction with RNA, thus resulting in a looser network formation, condensates that can remodel at the membrane surface, and RNA that is more accessible during replication steps. In its unmodified form, N binds tightly to RNA, and facilitates its encapsulation after being anchored to the membrane surface by N–M interactions. Through this model, we suggest that phase separation is critical to SARS-CoV-2 replication and that phosphorylation is a key mechanism through which phase separation is regulated to allow N to complete its functions at membrane surfaces.

Our results are distinct from recently published data on the effect of phosphorylation on the molecular conformation of N, determined using NMR. Botova et al. found a rigidification of the SR region upon phosphorylation that their data suggests is a result of new interactions with the RNA-binding domain[73]. Their conclusions are due to the similarity in the chemical shift perturbations observed when N is phosphorylated vs. when RNA binding occurs. However, the rigidification of the SR region could also be due to the presence of bulky phosphate groups. Though new interactions between N's RNA-binding domain and the linker region could be overlooked by our SAXS-derived reconstruction, given its low resolution, our results point to a larger conformational shift of the dimer. Stuwe et al. also studied the effect of phosphorylation on N using NMR, though the construct studied lacked the N-terminal RNA binding domain, thus potentially skewing the analysis of intra- and inter-molecular interactions[74]. They found that phosphorylation changes the behavior of a leucine-rich helix (LRH) found in N's linker: in unmodified N, higher-order oligomers form through LRH self-interaction that is significantly weakened by phosphorylation. In contrast with these results, our data presents an alternative model in which phosphorylation of N promotes intermolecular interactions between the SR regions of monomers within a

dimer and thus disrupts nonspecific interaction between pN and unstructured RNA. We do not observe a significant change in the oligomerization behavior of N following phosphorylation (Supplementary Fig. 14).

Our findings suggest several pathways towards the development of novel COVID therapeutics. One is to target the relevant kinases. A recent study suggested that N in cells is sequentially phosphorylated by host kinases SRPK1, GSK-3, and CK1[17]. Indeed, inhibiting SRPK1 was found to reduce viral replication, while blocking GSK-3 activity decreased replication in cells and reduced infection in patients[17,75]. Second, our results suggest that a population of unmodified N protein is necessary for viral encapsulation and formation of new virions. Although the source of this population of unmodified N is less understood, a phosphatase was identified to act on the murine coronavirus N protein[76]. Similar phosphatases should be identified for the SARS-CoV-2 N protein and explored as treatment targets. Third, our results also highlight that phase separation may be critical for the functions of N protein, and particularly that the material properties of condensates can support N's functions. Therefore, condensate-modifying molecules, which may not directly interact with N but do partition into condensates and affect their properties, may also interrupt viral replication and should be explored as treatments. Fourth, an alternative approach to inhibiting viral replication is to target RNA structure. We identified that RNA structure is important in determining the material properties of N and RNA condensates, and as such, molecules that disrupt viral RNA structure may be potential treatments for viral infection. For example, Vögele et al. identified a molecule that disrupts the double-stranded structure of stem loop 4 (SL4) in viral RNA[77]. Further investigation is needed to assess whether this type of disruption modulates the material properties of N and RNA condensates in a manner and extent that impairs N's functions in viral replication.

Although significant progress has been made in understanding the mechanisms of SARS-CoV-2 replication in infected cells, here we turned to an in vitro system to investigate the biophysical mechanisms driving the behaviors observed in cells. Where possible, we opted to include components in our system that mimic physiological conditions, such as in vitro phosphorylated N instead of a phosphomimetic version, and fragments of viral RNA instead of unstructured RNA. Using an in vitro system has allowed us to investigate mechanisms that would be challenging to measure in cells, such as quantifying how specific membrane protein domains affect condensate-membrane interactions, how specific RNA structures affect the strength of N protein binding, and how phosphorylation affects the conformation of N protein. However, we recognize that within the crowded environment of the cell, and with many other interaction partners present during viral infection, the behaviors of N may be different than those we studied in vitro. For example, phosphorylation of N protein has been shown to promote its interaction with 14-3-3 proteins, which are abundant in cells[78,79]. In our system, we did not consider how binding to 14-3-3 proteins may change how pN interacts with other viral membrane proteins and membranes. Therefore, further studies in infected cell models are necessary to verify our findings. For example, in Scherer et al., the location of pN was identified as the surface of replication organelles by studying infected cells that were fixed and stained. This experiment should be repeated with cells that are treated with SRPK1/2 inhibitors to verify how inhibition of phosphorylation affects the localization of N.

In conclusion, we have investigated the connection between the N protein's molecular conformation, its material properties when phase separated with RNA, and its membrane-associated behavior. We show that phosphorylation may act as a switch to toggle N between its two membrane-associated states during viral replication. Given the complexity of the system studied, further investigations are needed to better understand how our results apply to other N

protein interactions that occur in infected cells. This includes considering factors such as the complex composition of lipid membranes, as well as the presence of both human mRNA and the much longer viral genomic RNA.

## Methods

### Cloning
All genes of interest were cloned into pET vectors in frame with N-terminal 6xHis tags. A TEV protease site was inserted between the 6xHis tag and the Nucleocapsid protein coding sequence. Constructs were cloned by DNA assembly (NEBuilder HiFi DNA Assembly Master Mix; New England Biolabs). Gene sequences were verified by Sanger sequencing (Genewiz).

### Protein expression and purification
For bacterial expression, plasmids were transformed into BL21(DE3) competent *E. coli* (New England BioLabs). Colonies picked from fresh plates were grown for 12 h at 37 °C in 5 mL LB while shaking at 250 rpm. This starter culture was then used to inoculate 0.5 L cultures. Cultures were grown overnight in 2 L baffled flasks in Terrific Broth medium (Fisher Scientific) supplemented with 4 g/L glycerol at 18 °C while shaking at 250 rpm. Once the $OD_{600}$ reached -1, expression was induced with 500 μM isopropyl β-ᴅ-1-thiogalactopyranoside (IPTG). The pET vectors used contained a kanamycin resistance gene; kanamycin was used at concentrations of 50 μg/mL in cultures[80]. After overnight expression at 18 °C, bacterial cells were pelleted by centrifugation at 4100 × $g$ at 10 °C. Pellets were resuspended in lysis buffer (1 M NaCl, 20 mM Tris, 20 mM imidazole, EDTA-free protease inhibitor, and pH 7.5) and lysed by sonication while on ice. Lysate was clarified by centrifugation at 25,000 × $g$ for 30 min at 10 °C. The clarified lysate was then filtered with a 0.22 μm filter.

Proteins were purified using an AKTA Pure FPLC with 1 mL nickel-charged HisTrap columns (Cytiva) for affinity chromatography of the His-tagged proteins. For N protein, after injecting proteins onto the column, the column was washed with 15 column volumes of 3 M NaCl, 20 mM Tris, 20 mM imidazole, and pH 7.5. For all other proteins, after injecting proteins onto the column, the column was washed with 5 column volumes of 500 mM NaCl, 20 mM Tris, 20 mM imidazole, and pH 7.5. Proteins were eluted with a linear gradient up to 500 mM NaCl, 20 mM Tris, 500 mM imidazole, and pH 7.5. Histidine tags were cleaved from the N protein using TEV protease during dialysis. TEV protease was added to N protein and dialyzed overnight using 10 kDa MWCO membranes (Slide-A-Lyzer G2, Thermo Fisher) into 300 mM NaCl, 20 mM Tris, 20 mM imidazole, 5 mM DTT, pH 7.5 buffer at 4 °C. The reaction mixture was purified using a nickel resin gravity column (HisPur Ni-NTA Resin, Thermo Fisher) and the flow-through was collected. Flow-through aliquots were concentrated and buffer exchanged into storage buffer (300 mM NaCl, 20 mM Tris-HCl, and pH 7.5) using a 10 kDa MWCO centrifugal filter (Amicon Ultra, Sigma). Proteins were either reserved for the phosphorylation protocol or snap frozen in liquid $N_2$ in single-use aliquots and stored at -80 °C.

### SDS-PAGE
For chromatographically purified proteins, SDS-PAGE was run using NuPAGE 4–12% Bis-Tris gels (Invitrogen) and stained using a Coomassie stain (GelCode Blue Safe Protein, Thermo Scientific).

### Fluorescent labeling of the N protein
Purified N protein was dialyzed into PBS buffer with 0.1 mL sodium bicarbonate per 1 mL protein solution. Protein was then labeled by adding a 3:1 molar ratio of Alexa-647 NHS ester (Fisher Scientific) stored in DMSO. The mixture was incubated at 4 °C for 1 h with rocking. Unbound dye was removed by size exclusion chromatography

(Superdex 200 Increase 10/300 GL, Cytiva). For phase separation assays, the percent of dyed protein was adjusted to 5% of the total by dilution with undyed protein.

## Phosphorylation protocol

Approximately 80 µM N protein was prepared in a buffer containing 300 mM NaCl, 20 mM Tris, 1 mM dithiothreitol (DTT), 10 mM MgCl$_2$, and 2 mM ATP in a 200 µL reaction mixture. About 5 µL GSK-3β and 5 µL SRPK (SignalChem Biotech) were added to the mixture. After incubation at 37 °C for 120 min, phosphorylation was confirmed using a SuperSep Phos-tag acrylamide gel (FujiFilm Wako Chemicals). Enzymes were removed from the reaction mixture using GST-based affinity chromatography. Briefly, 1 mL of glutathione resin (Glutathione Sepharose 4B, Cytiva) was packed in a gravity column. The column was washed with 300 mM NaCl, 20 mM Tris, and pH 7.5 buffer. The 200 µL of reaction mixture was diluted to a total of 2 mL using wash buffer and poured into the column. The column flow-through was collected. The column was washed with an additional 1 mL of wash buffer, which was collected. The flow-through and wash fractions were combined, and the 3 mL of solution containing phosphorylated N protein was concentrated and buffer exchanged using an Amicon Ultra 0.5 mL 10 K MWCO centrifugal filter, according to manufacturer instructions.

## RNA in vitro transcription protocol

RNA production was carried out according to established protocols[81]. Templates were gifted from the Gladfelter lab, where they were synthesized (IDT) and cloned into pJet (Thermo Fisher Scientific K1231) plasmids using blunt-end cloning. Directionality and sequence were confirmed using Sanger sequencing (Azenta). Plasmids were linearized and then amplified using PCR. About 10 ng of plasmid was used as starting material, and primers are noted in the Supplemental Materials. Melting temperatures for the 1–1000 and N fragments were 69 °C. About 5 µL of PCR product was loaded onto an agarose gel to determine size and purity using SYBR™ Gold Nucleic Acid Gel Stain (Invitrogen) and 1 kb Plus DNA Ladder (New England Biolabs). If the PCR product was pure, then the sample was purified (NEB DNA Clean-up kit). About 100 ng of purified DNA was used as a template for in vitro transcription (New England Biolabs), carried out according to the manufacturer's instructions. Following incubation at 37 °C for 18 h, in vitro transcription reactions were treated with DNAse (New England Biolabs) according to the manufacturer's instructions. Following DNAse treatment, reactions were purified with an RNA purification kit (NEB T2040L). Purified RNA was verified for purity and size using an agarose gel and RNA Gel Loading Dye (NEB B0363S) and RiboRuler High Range RNA Ladder (Thermo Scientific SM183). Concentration was measured using a Nanodrop One spectrophotometer (Thermo Fisher Scientific).

Fluorescently labeled RNA was gifted by the Gladfelter lab. Cy3 RNA was transcribed from the same template used above and using the same protocol described above, but with the addition of 0.1 µL of Cy3 labeled UTP to each reaction (Sigma PA53026).

## Microscopy

For microscopy experiments, protein samples were prepared as follows: N or pN protein aliquots were thawed at room temperature. Proteins were then diluted and mixed with RNA (if any) to obtain a final solution with 40 µM protein and 300 nM viral RNA or 1 mg/mL polyrA (unless otherwise noted) and buffer conditions of 20 mM Tris-HCl, 150 mM NaCl, and pH 7.5. Protein concentrations were measured based on their absorbance at 280 nm using a Nanodrop spectrophotometer (Thermo Fisher).

Protein samples were plated on 1.5 thickness slides that were coated with 5% Pluronic F-127 (Sigma-Aldrich) for a minimum of

10 min. The slides were washed with buffer solution prior to plating the protein samples. A silicone spacer (0.5 mm) and a microfluidic temperature controller (Cherry Biotech) were attached to the slide and the temperature of the sample was set to 37 °C during observation.

Confocal imaging was performed on a Zeiss Axio Observer 7 inverted microscope equipped with an LSM900 laser scanning confocal module and employing a 63x/1.4 NA plan-apochromatic, oil-immersion objective. GFP was excited to fluoresce with a 488 nm laser, Cy3 with a 561 nm laser, and Alexa-647 with a 640 nm laser. Confocal fluorescence images were captured using GaAsP detectors. Transmitted light images were collected with either the ESID module or an Axiocam 702 sCMOS camera (Zeiss), in both cases using a 0.55 NA condenser.

## Droplet image analysis

Image analysis and data processing for Figs. 2, 3, 4a–c and Supplementary Figs. 2, 3, 4, 5, 10 were performed in MATLAB R2023a. The fluorescence intensity profile of the condensates with fluorescently labeled proteins was measured by using the Hough Transform to identify droplet locations and drawing a line that spanned the droplet diameter plus 1/4th of a radius length in each direction across the droplets. Line-scan graphs were generated in MATLAB. Total intensity and partitioning graphs were generated in MATLAB. Condensate perimeter and area were calculated using MATLAB's inbuilt region-props function.

## Giant unilamellar vesicle electroformation

Giant unilamellar vesicles were prepared by electroformation[82]. Briefly, 20 µL of a 10 mM lipid solution containing 60% DOPC, 25% DOPE, 10% DOPS, 5% DGS-NTA(Ni) in chloroform (Avanti Lipids) was prepared. For experiments including cholesterol, 2%, 5%, or 25% cholesterol was included in place of the corresponding % of DOPC. Approximately 10 µL lipid solution was spread onto indium tin oxide (ITO)-coated glasses and dried under vacuum for 12 h. The plates were assembled into a chamber with a Teflon spacer, and the swelling solution (~500 µL of 300 mM sucrose) was introduced. For electroformation, a sinusoidal electric field of 2.0 V peak-to-peak and 10 Hz was applied using a function generator for 2.5 h, after which the frequency was reduced to 5 Hz for 30 min. Osmolarity of the sucrose solution (301 ± 1 mOsM) was lower than the osmolarity of the condensate NaCl solutions (324 ± 1 mOsM) to ensure vesicles were not too tense. The solution osmolarities were confirmed using a freezing-point osmometer (Advanced Instruments model 3D3).

## Contact angles measurement and geometric factor calculation

A detailed explanation of the contact angles used in this work has been published elsewhere[37]. Briefly, we measured the three contact angles $\theta_c$, $\theta_e$, $\theta_i$ from fluorescent images of GUVs and condensates (for samples with GFP) or from brightfield images (for samples without GFP). From these angles, we calculated the geometric factor, $\Phi$. From Mangiarotti et al., 2024, when $\Phi = -1$ there is complete wetting of the membrane by the condensate phase, while $\Phi = +1$ corresponds to dewetting of the membrane by the condensate phase. The geometric factor, $\Phi$, is negative if the membrane prefers the condensate over the exterior buffer and positive otherwise[37]. We followed the same procedure to measure $\theta_{in}$, the intrinsic contact angle.

## Fluorescence recovery after photobleaching

Circular bleach regions of approximate radius $R = 1$ µm were drawn in the center of protein droplets. Alexa-647 was imaged and bleached with a 640 nm laser. Cy3 was imaged and bleached with a 561 nm laser.

Recovery curves were fit to a single exponential recovery model

$$f(t) = A^*\left(1 - e^{-t/\tau}\right) \tag{1}$$

to calculate the recovery timescale, $\tau$.

## Passive microrheology

Yellow–green carboxylate-modified polystyrene beads (0.5 μm diameter; FluoSpheres, Invitrogen) were used for video particle-tracking (VPT) microrheology measurements. Each sample was prepared by mixing protein and RNA to a final concentration of 40 μM protein and 300 nM viral RNA or 1 mg/mL polyrA in 150 mM NaCl, 20 mM Tris-HCl buffer, pH 7.5. Next, 50 μL of the protein and RNA sample was mixed with 1 μL of a 1:100 dilution of the fluorescent tracer bead solution and the sample was plated on a 1.5 thickness slide pretreated with Pluronic F-127. The sample was covered with a CherryTemp microfluidic temperature controller chip, and the sample was set to 37 °C and incubated for 1 h.

VPT measurements were conducted on a Zeiss Axio Observer 7 inverted microscope equipped with an Axiocam 702 monochrome sCMOS camera (Zeiss), employing a 63x, 1.4-numerical aperture plan-apochromatic oil-immersion objective. The microscope focus was adjusted to the midsection of the protein droplets for VPT acquisition. Epifluorescence video imaging was initiated at the 1-h timepoint, with fluorescence excitation using a 475-nm light-emitting diode (Colibri 7, Zeiss). Videos of the tracer beads diffusing within the condensate were collected at 100 frames per second for 2000 frames. Imaging was conducted at 37 °C. For each sample, three independent samples were made on different days, and ~20 videos were collected from each sample, with each video containing ~5–50 tracer beads. Viscoelasticity data presented in Fig. 4d–h and Supplementary Fig. 10E–F are the average of these independent trials.

Data analysis was conducted using the open-source particle tracking package TrackPy (v0.5.0)[83] in Python and customized as needed. The TrackPy particle tracking code was used to analyze the collected videos, starting with extracting particle trajectories. The MSD was calculated from the trajectories of individual beads, followed by calculating the ensemble-average MSD. To remove the static error from the MSD curves for calculating viscosities, we corrected the ensemble-average MSD by subtracting the noise floor from the MSD curves[84]. In general, the ensemble-average MSD often scales as a power law with lag time τ, as given by

$$<r^2(\tau)> = 2dD\tau^{\alpha} \tag{2}$$

where d is the number of dimensions (here d = 2, because data collection and analysis were conducted in the x–y plane), D is the diffusion coefficient, and α is the diffusivity exponent. For a purely viscous fluid, the diffusivity exponent α is unity, and the Stokes-Einstein relation can be used to calculate the viscosity[85]. The α values for all the condensates tested were in the range of 0.3–1.1. We therefore used the generalized Stokes-Einstein relation (GSER) to measure the viscoelastic properties of the condensates[85]. The frequency-dependent GSER in the Fourier domain is represented by the following equation:

$$G^*(\omega) = G'(\omega) + iG''(\omega) = \frac{dk_BT}{3\pi a(i\omega)<\Delta r^2(\omega)>} \tag{3}$$

where G*(ω) is complex shear modulus, $k_B$ is Boltzmann's constant, T is the temperature, a is the bead radius, G'(ω) and G''(ω) are the frequency-dependent storage (elastic) and loss (viscous) moduli, respectively, and $<\Delta r^2(\omega)>$ is the unilateral Fourier transform of the MSD. We use an algebraic approach proposed by Mason et al. to

estimate the Fourier transform of the MSD, which approximates the local MSD as a power-law function[56]. The algebraic expression is given by the following equations:

$$|G^*(\omega)| = \frac{dk_BT}{3\pi a < \Delta r^2\left(\tau = 1/\omega\right) > \Gamma\left[1 + \alpha\left(\tau = 1/\omega\right)\right]} \tag{4}$$

$$G'(\omega) = |G^*(\omega)| \cos\left(\frac{\pi\alpha(\omega)}{2}\right) \tag{5}$$

$$G''(\omega) = |G^*(\omega)| \sin\left(\frac{\pi\alpha(\omega)}{2}\right) \tag{6}$$

where $\alpha(\tau) = d\ln(\Delta r^2(\tau))/d\ln(\tau)$ is the local power law exponent describing the logarithmic slope of $\Delta r^2(\tau)$ at $\tau = 1/\omega$ and Γ is the Gamma function. Next, cubic spline interpolation fitting is used on the calculated G' and G'' data to reduce measurement noise generated in the algebraic conversion[86]. We plot the fitted G' and G'' to identify the crossover frequency, or the timescale at which the material transitions from an elastic-dominant to a viscous-dominant regime. To extract the zero-shear viscosity from the viscoelastic moduli, we use the following equation:

$$\eta(\omega) = \frac{G''(\omega)}{\omega} \tag{7}$$

and obtain the limit of η(ω) at low frequencies[56]. The presented viscoelastic moduli and zero-shear viscosities are the average from multiple videos (n = 10) taken from three independent samples.

The noise floor of the 500 nm beads was measured by allowing a solution of beads in water to dry on the glass surface of a slide, resulting in beads adhering to the glass surface. We acquired the trajectories of the beads adhered to the glass surface using the same parameters and experimental setup as used for VPT studies of the samples.

## Micropipette aspiration

The micropipette aspiration experiments were carried out on a Ti2-A inverted fluorescence microscope (Nikon, Japan) equipped with a motorized stage and two motorized 4-axes micromanipulators (PatchPro-5000, Scientifica) and a multi-trap optical tweezers (Tweez305, Aresis, Slovenia) according to the protocol we reported previously[58,59]. An oil-immersion objective (100X; NA 1.30; Nikon) was integrated with an objective heating collar (OKOlab) and temperature controller (OKOlab) for 37 °C measurements. Micropipettes were made by pulling glass capillaries using a pipette puller (PUL-1000, World Precision Instruments). The pipette tip was then cut to achieve an opening diameter of ~5 μm. Subsequently, the pipette was bent to an angle of approximately 40° using a microforge (DMF1000, World Precision Instruments).

MPA experiments were carried out in glass-bottom dishes (ES56291, Azer Scientific, USA) that were pretreated with 5% Pluronic F-127 (P2443-250G, Sigma) for 10 min to prevent adhesion of condensates to the glass. The micropipette was filled with the same buffer used in microscopy experiments (150 mM NaCl, 20 mM Tris-HCl, and pH 7.5) using a MICROFIL needle (World Precision Instruments). The filled micropipette was then mounted onto a micromanipulator. The rear end of the pipette was connected to an automatic pressure controller (Flow-EZ, Fluigent; Pressure resolution 1 Pa).

Optical tweezers were used to contact and merge droplets to achieve a large (>5 μm) condensate for easier MPA measurements and analysis. A secondary micropipette was used to hold the condensate during MPA. To minimize sample evaporation, 1.5 mL Milli-Q water was added to the edge of the dish, and the dishes were covered with a thin

plastic wrap with a ~2 mm hole for pipette insertion. We observed that in vitro condensates always wet the inner wall of uncoated micropipettes. Therefore, the analysis of the MPA data follows the protocol described in ref. 59. Briefly, normalized aspiration length (aspiration length, $L_p$, over the pipette radius, $R_p$) was segmented according to the pressure steps. For each segment, the slope of a linear fitting of $(L_p/R_p)^2$ vs. time is equal to the effective shear rate. Then, the slope of the aspiration pressure vs. shear rate graph gives $4\eta$, where $\eta$ is the condensate's viscosity.

## Fluorescence polarization

Fluorescence polarization measurements were performed on a Tecan Spark microplate reader in a 384-well black plate at 20 °C. A monochromator set the excitation wavelength at 485 nM and the emission wavelength at 535 nm, with a 20 nm bandwidth. Purified N or pN protein was serially titrated in 150 mM NaCl, 20 mM Tris·HCl, and pH 7.5 buffer and incubated with a constant 4 nM FAM-labeled RNA (Azenta) for 10 min at room temperature (20 °C) prior to measurement.

The data were analyzed using MATLAB V2023a, with a one-site binding curve (hyperbola) fitted to the data. $K_D$ values from three experiments were averaged, and the standard deviation was calculated. Data were normalized by subtracting the initial polarization value from each dataset. The one-site binding equation utilized is

$$mP = \frac{mP_{Max}{}^*[N]}{K_D + [N]} \quad (8)$$

in which mP is the observed millipolarization, $mP_{MAX}$ is the maximum polarization, [N] is the concentration of protein (unmodified or phosphorylated).

## smFRET

Single-molecule fluorescence measurements were performed as described in ref. 8 with a Picoquant MT200 instrument (Picoquant, Germany). Briefly, FRET experiments were performed by exciting the donor dye with a laser power of ~100 μW (measured at the back aperture of the objective) at 488 nm wavelength. For pulsed interleaved excitation of donor and acceptor, we used a repetition rate of 20 MHz for the donor excitation and a delay of ~25 ns for acceptor excitation. Acceptor excitation was achieved by using a white laser (SuperK Extreme, NKT Photonics, Denmark), filtered by a z582/15 band pass filter (Chroma). Acceptor excitation power was adjusted to match the acceptor emission intensity to that of the donor (between 50 and 70 μW). Single-molecule FRET efficiency histograms were acquired from samples with protein concentrations of 100 pM labeled protein, and the population with stoichiometry corresponding to 1:1 donor:acceptor labeling was selected. Photon detection events were stored with 16 ps resolution.

Proteins were designed and prepared as described in ref. 8. Following preparation, protein for smFRET experiments was phosphorylated using the protocol described above, with the exception of the use of 200 mM β-mercaptoethanol instead of 5 mM DTT. All samples were prepared in 50 mM HEPES pH 7.4, 150 mM KCl, 200 mM β-mercaptoethanol (for photoprotection), 0.001% Tween 20 (for limiting surface adhesion) unless otherwise stated. All measurements were performed in custom-made glass cuvettes coated with PEG. Each sample was measured for at least 30 min at room temperature (295 ± 0.5 K). Protein concentrations were (1) 100 pM labeled protein for the low concentrations, (2) 100 pM labeled protein + 1 μM unlabeled protein for high concentration for the full-length construct, or (3) 100 pM labeled protein + 4 μM unlabeled protein for high concentration for $N^{1-246}$.

Determination of root mean square inter-dye distances from mean FRET transfer efficiencies was conducted as described in ref. 87. Briefly, the Gaussian chain model was employed in the conversion, which relies on a single parameter, the root mean squared inter-dye distance $r = <R^2>^{1/2}$

$^2$. Estimates for this parameter were obtained by numerically solving:

$$\langle E \rangle = \int_0^\infty E(R)P(R)dR \quad (9)$$

where <E> is the mean transfer efficiency, $R$ is the inter-dye distance, $P(R)$ represents the Gaussian chain distribution, and $E(R)$ is the Förster equation for the dependence of transfer efficiency on distance $R$ and Förster radius $R_0$.

## smFRET lifetime analysis

Comparison of transfer efficiency and fluorescence lifetime indicates whether the transfer efficiency reports about a rigid distance (e.g., structure formation or persistent interaction with a structure) or is the result of a dynamic average across multiple conformations. Analysis of fluorescence lifetime from mean FRET transfer efficiencies was conducted as described in ref. 66. We compared how fluorescence lifetimes depend on transfer efficiencies for each burst with the behavior expected for fixed distances and for a chain sampling a broad distribution of distances. For a fixed distance, R, the mean donor lifetime in the presence of an acceptor is given by:

$$t_D(R) = t_{D0}(1 - E(R)) \quad (10)$$

where $t_D$ is the lifetime in the absence of acceptor, and

$$E(R) = \frac{1}{1 + R^6/R_0^6} \quad (11)$$

For a chain with a dye-to-dye distance distribution P(R), the donor lifetime is

$$t_D = \frac{\int t I(t)dt}{\int I(t)dt} \quad (12)$$

$$I(t) = I_0 P(R) Exp[-t/t_D(R)]dR \quad (13)$$

Where I(t) is the time-resolved fluorescence emission intensity following donor excitation. Donor and acceptor lifetimes were analyzed by fitting subpopulation-specific time-correlated photon counting histograms after donor and acceptor excitation, respectively, using a tail fit. Errors associated with the tail fit are estimated by varying the "tail" region that undergoes the fitting procedure and computing the mean and standard deviation of the fit results. In computing the average of multiple measurements, errors of the single dataset are propagated accordingly.

## smFRET dimerization constant

Modeling of dimerization was conducted as described in ref. 66. Binding experiments were analyzed, accounting for the possibility of forming dimers of labeled molecules $DD_L$ with other labeled molecules ($DD_L$:$DD_L$), labeled molecules with other unlabeled molecules $DD_U$ ($DD_L$:$DD_U$), as well as between unlabeled molecules ($DD_U$:$DD_U$). Each dimer then has the corresponding dissociation constant:

$$K_D^{LL} = \frac{[DD_L]^2}{[DD_L : DD_L]} \quad (14)$$

$$K_D^{UU} = \frac{[DD_U]^2}{[DD_U : DD_U]} \quad (15)$$

$$K_D^{LU} = \frac{[DD_L][DD_U]}{[DD_L : DD_U]} \quad (16)$$

Finally, we accounted for the fact that if expressed in the microscopic rate constants of binding and unbinding the $K_D^{LU} = 1/2K_D^{UU} = 1/2K_D^{LL}$, under the assumptions that the rates of unlabeled and labeled are equal. The fraction of labeled protein that forms dimers with unlabeled protein (coinciding with the FRET population at 0.5 stoichiometry ratio) is given by:

$$f_{Dimer}^{LU} = \frac{[DD_u]^{tot}(K_D^{LU} + 2\left([DD_L]^{tot} + [DD_u]^{tot}\right) - \sqrt{K_D^{LU}}\sqrt{k_D^{LU} + 4([DD_L]^{tot} + [DD_U]^{tot})}}{2\left([DD_L]^{tot} + [DD_U]^{tot}\right)^2}$$

(17)

## Small-angle X-ray scattering

Small-angle X-ray scattering (SAXS) measurements were made at 16-ID-C LIXS beamline (National Synchrotron Light Source II (NSLS-II), Brookhaven National Laboratory; 15.14 keV X-rays ($\lambda = 0.8189$ Å) and two Pilatus 1 M detectors). Samples were prepared by diluting protein into 150 mM NaCl, 20 mM Tris-HCl buffer to the concentrations desired (0.5 mg/mL–4 mg/mL). Data over a $q$ range of 0.005–0.25 Å$^{-1}$ was analyzed. Background subtraction was done using the scattering from the storage buffer (150 mM NaCl, 20 mM Tris-HCl, and pH 7.5) and by scaling the buffer and sample intensities at the $q \sim 2$ Å$^{-1}$ water peak.

The data were analyzed in BioXTAS RAW 2.1 with ATSAS 3.0.4 to determine the radius of gyration ($R_g$) by Guinier analysis, the compactness of the particle by Kratky plots and pair distance distribution functions, $P(r)$. $R_g$ from $P(r)$ was compared with the Guinier $R_g$ to ensure internal consistency in data analyses. $D_{max}$ was calculated from $P(r)$. Bead model reconstructions using a dummy atom model were obtained from the $P(r)$ functions generated and stored as .out files using GNOM in BioXTAS RAW 2.1 using the SAXS data as input. Bead models were generated using the DAMMIN program in the ATSAS 3.0.4 software package, assuming single-phase objects.

## LC-MS

**Whole protein.** Protein sample ($n = 1$) was diluted to 1 mg/ml with 50 mM ammonium acetate, pH 6 and analyzed by LC-MS using Vanquish UPLC coupled to QExactive HF mass spectrometer (Thermo Fisher). About 1 μg of sample was injected onto a desalting cartridge (C18, 2 mm diameter, Phenomenex security guard) with a flow rate of 200 μl/min. Gradient: 2–8 min from 15 to 90% B (A: 0.2% Formic acid, B: 0.1% Formic acid, Acetonitrile). The eluate was ionized in the HESI source with sheath gas nitrogen flow rate at 25, at ambient temperature with spray voltage of 4 kV. The in-source decay of 10 V was applied. The S-lens RF was set at 70. The AGC was set at $3 \times 10^6$ and the max ion time was set at 200 ms. MS was acquired in the range of m/z 500−3000 in profile mode with summed 5 micro scans at a resolution of 15k. Protein peak was summed, and the resulting spectrum was deconvolved with UniDec 7.0.1[88] with m/z set at 600 to max, charge states from 1 to 50, mass range from 1 K to 500 K Da, and sample mass every 1.0 Da.

**Pepsin digest.** For pepsin digestion, a protein sample ($n = 1$) was prepared with 2 μL of protein and 38 μL PBS and 60 μL of denaturation buffer (2 M urea, 20 mM TCEP, 0.8% formic acid). Sample was injected into a Vanquish LC system coupled to a Q Exactive HF mass spectrometer and digested online using a protease type XVIII/pepsin column (NovaBioassays). Peptides were first trapped on a Waters BEH C18 trap column (2.1 mm × 2 cm), then separated on a Hypersil Gold C18 analytical column (2.1 mm × 5 cm, 1.9 μm particle size) using a multi-step gradient at 5 °C: 0–4 min at 2% B, ramping to 10% B in 0.3 min, followed by 10 to 40% B over 7.6 min. MS data were acquired at 120,000 resolution over a 300–2000 m/z range. The top 20 most intense precursor

ions were selected for fragmentation using HCD (collision energy 27), with MS/MS scans acquired at 15,000 resolution and a dynamic exclusion of 20 s. Data were analyzed in Proteome Discoverer 3.0 using Sequest HT against a custom database containing target proteins, kinases, and common lab contaminants. Precursor and fragment mass tolerances were set to ±10 ppm and ±0.02 Da, respectively. N-terminal acetylation was set as a dynamic protein terminus modification, while methionine oxidation and phosphorylation at serine, threonine, and tyrosine were set as dynamic modifications. A nonspecific protease setting was used. Peptide-spectrum matches were validated using the Fixed Value PSM Validator (maximum delta CN <0.05), and phosphosite confidence was assessed with PhosphoRS.

**Trypsin digest.** Tryptic digest was prepared using the single-pot solid-phase-enhanced sample preparation (SP3) method with 10 μg of protein per sample ($n = 1$). Following protein binding and washing, digestion was initiated, and after 30 min at 37 °C, 20 μL was removed, acidified with 1 μL of 10% formic acid, and stored at −80 °C. The remaining sample was incubated overnight at 37 °C to complete digestion. Peptides from each timepoint were prepared for LC-MS/MS by mixing 10 μL of digest with 10 μL of 0.1% TFA (1:1), resulting in a final concentration of 0.1 μg/μL. From this, 5 μL (0.5 μg) was injected into a Q Exactive HF mass spectrometer using a 70-min data-dependent acquisition (DDA) method. MS/MS data were searched against a custom protein database using MSFragger.

## BS3 cross-linking experiment

Protein (2–10 μM in PBS at pH 7.5) was chemically cross-linked with bis(sulfosuccinimidyl) suberate (BS3) (Thermo Fisher) (50 μM) at room temperature for 30 min and quenched by adding Tris (pH 7.5) to a 20 mM final concentration. Reaction mixtures were then analyzed by SDS-PAGE.

## Molecular dynamics simulations

All molecular dynamics (MD) simulations were performed using OpenMM 7.6.0[89] in the NVT ensemble. Trajectory analyses were performed using the MDAnalysis Python library (v2.4.3), and figures were generated using Matplotlib (v3.8.4). We simulated single truncated monomers, or full dimers of the N-protein, using Langevin dynamics at a constant temperature of 310 K, with a friction coefficient of 0.01 ps$^{-1}$ and an integration time step of 10 fs. Each system was enclosed in a cubic periodic box with periodic boundary conditions. Each chain was simulated for 260 ns, including a 10 ns equilibration phase followed by 250 ns of production dynamics. From the resulting trajectories, the first 50 ns of simulation were discarded as additional equilibration time before analysis. For simulations of the full dimer, since we sample total conformational rearrangements and observe many transitions from collapsed to extended, we conclude that the results are effectively independent of the initial configuration, as the system explores the relevant ensemble of states within the simulation timescale.

Force field and interaction potentials: we employed the HPS-Urry coarse-grained energy model[69], which includes bonded interactions, non-bonded interactions, and an elastic network model (ENM) for structured regions. The total interaction energy U$_{total}$ is given by:

$$U_{total} = \sum_{i>j} \phi_{ij}^{elec} + \sum_{i>j} \phi_{ij}^{HPS} + \sum_{i=1}^{N-1} k_b(r_{i,i+1} - r_0)^2 + \sum_{(i,j) \in pairs} \frac{1}{2}k_d(r_{ij} - r_0^{(ij)})^2$$

(18)

where $\phi_{ij}^{elec}$ represents electrostatic interactions, $\phi_{ij}^{HPS}$ denotes hydrophobic interactions, and the third term models bonded interactions between adjacent residues using a spring constant $k_b = 20 kj/A^2$ and an equilibrium bond length r$_0 = 3.82 A$. The last

term corresponds to the elastic network model (ENM), which restrains the folded domains and is defined below.

Hydrophobic interactions are modeled using the Ashbaugh–Hatch potential[90]:

$$\phi_{ij}(r) = \begin{cases} \phi_{ij}^{LJ}(r) + \left(1 - \lambda_{ij}\right)\epsilon & ,for\ r \leq 2^{1/6}\sigma_{ij}, \\ \lambda_{ij}\phi_{ij}^{LJ}(r), & for > 2^{1/6}\sigma_{ij} \end{cases} \tag{19}$$

with the Lennard-Jones potential defined as:

$$\phi_{ij}^{LJ}(r) = 4\epsilon\left[\left(\frac{\sigma_{ij}}{r}\right)^{12} - \left(\frac{\sigma_{ij}}{r}\right)^{6}\right] \tag{20}$$

In this, $\epsilon = 0.2$ kcal/mol. $\lambda_{ij}$ and $\sigma_{ij}$ can be calculated by the arithmetic average of hydrophobicity values $((\lambda_{ij} = \lambda_i + \lambda_j)/2)$ and size $((\sigma_{ij} = \sigma_i + \sigma_j)/2)$, where $\sigma_i$ and $\lambda_i$ are the size and stickiness, respectively, of each amino acid type. Values of $\lambda$ come from the Urry hydropathy scale[69,91], and $\sigma$ from the predicted volume of amino acid residues[92].

Electrostatics are modeled using a Debye–Hückel potential[93]:

$$\phi_{ij}^{el}(r) = \frac{q_i q_j}{4\pi D r} e^{-\kappa r} \tag{21}$$

where $q_i$ represents the charge of residue "i" located at the center of the corresponding bead. D is the dielectric constant of the solvent, and $\kappa$ is the inverse Debye screening length. For our calculations, we set $D = 80$, the dielectric constant of water, and $\kappa = 1\,nm^{-1}$, which corresponds to salt concentrations of ~100 mM.

Elastic network model (ENM) for folded domains: to maintain structural integrity of folded domains, we applied an elastic network model (ENM) using a harmonic potential between non-bonded residue pairs within a cutoff distance based on the reference PDB structure[94]. The ENM potential is defined as:

$$u_{ENM}(r) = \sum_{(i,j)\in pairs} \frac{1}{2} k_d \left(r_{ij} - r_0^{(ij)}\right)^2 \tag{22}$$

where $r_{ij}$ is the current distance between two beads, $r_0^{ij}$ is the reference distance between residues i and j from the crystal structure, and $k_d$ is the force constant. For structured domains, each bead was placed at the center of mass calculated from all atoms in a residue (COM representation), while for intrinsically disordered domains, each bead was placed at the $C_\alpha$ atom position as has been used in similar models[70]. This model ensures that secondary and tertiary structures within folded domains remain stable throughout the simulation. Non-bonded interactions, including Ashbaugh–Hatch and Debye–Hückel potentials, are excluded for pairs restrained by the ENM.

The pair distance distribution function (PDDF) P(r) for each model was calculated by finding the cartesian distance between all residue pairs i,j. Contact between residues was defined as any two residues that were less than 1 nm apart, while ignoring interactions with itself, and two subsequent residues in sequence. Finally, a Bayesian maximum entropy ensemble reweighting scheme was implemented to adjust the statistical weights of the simulated conformations according to ref. 71.

### Reporting summary

Further information on research design is available in the Nature Portfolio Reporting Summary linked to this article.

## Data availability

Unless otherwise stated, all data supporting the results of this study can be found in the article, supplementary, and source data files. Plasmids generated in this study were deposited to Addgene. The SAXS data generated in this study have been deposited in the SASBDB database under accession codes SASDXX5, SASDXY5, SASDXZ5, SASDX26, SASDX36, SASDX46, SASDX56, SASDX66, SASDX76, SASDX86, and SASDX96. The mass spectrometry data generated in this study have been deposited in the MassIVE database under accession code MSV000098426 [https://massive.ucsd.edu/ProteoSAFe/dataset.jsp?task=74accbaf6e59410daa7c06cfcef07228]. Source data are provided with this paper.

## Code availability

All the quantitative analyses discussed in this paper were generated based on data and computer codes that are available in the Zenodo database under favettabruna/SARS-CoV-2-Nucleocapsid-data: SARS-CoV-2 Nucleocapsid Protein Image Analysis [https://zenodo.org/records/15889195].

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

## Acknowledgements

We thank Christine Roden and Amy Gladfelter for helpful discussion, for sharing plasmids for making RNA fragments, and fluorescently labeled viral RNA; Mihai Solotchi and Smita Patel for assistance with fluores-cence polarization experiments and helpful discussions; Gabriela Tirado-Mansilla for assistance preparing SAXS samples; Bineet Sharma for help with GUV production; Srinivas Chakravartula for assistance with mass spectrometry; Kevin Corbett and Qiaozhen Ye for sharing the GFP-Mfragment plasmid; David Shreiber for allowing us to use his osm-ometer; and Jean Baum and Vikas Nanda for helpful discussion. This work was supported by NIH grants R35GM142903 (to B.S.S.), R35GM147027 (to Z.S.), R35GM138296 (to A.J.G.), R35GM150589 (to G.D.) and National Science Foundation grant DMREF-2118860 (to N.S.M. and A.J.G.). X-ray scattering measurements were done at the NSLS-II beamline 16-ID (LiX) at Brookhaven National Laboratory. The LiX beam-line is supported by an NIGMS P30 Grant (P30GM133893), NIH Grant S10 OD012331, the DOE Office of Biological and Environmental Research (KP1605010), and the DOE Office of Science, Office of Basic Energy Sciences Program under contract number DE-SC0012704 to CBMS. Schematics were created in BioRender. Figure 1c: Created in BioRender. Favetta, B. (2025) https://BioRender.com/speaa8j. Figure 1d: Created in BioRender. Favetta, B. (2025) https://BioRender.com/o21g999. Figure 2a: Created in BioRender. Favetta, B. (2025) https://BioRender.com/s97t568. Figure 2b: Created in BioRender. Favetta, B. (2025) https://BioRender.com/we3zbf2. Figure 2d and Supplementary Figs. 4a, 10: Created in BioRender. Favetta, B. (2025) https://BioRender.com/654vtk2. Figure 2f: Created in BioRender. Favetta, B. (2025) https://BioRender.com/j5wt6ys. Figure 3a: Created in BioRender. Favetta, B. (2025) https://BioRender.com/o28w195. Figure 7: Created in BioRender. Favetta, B. (2025) https://BioRender.com/382j936.

## Author contributions

B.F.: conceptualization, investigation, data curation, formal analysis, and methodology; H.W.: methodology and investigation for MPA experiments; J.C. and A.Soranno: methodology, investigation and formal analysis for smFRET experiments; A.Singh: investigation and formal analysis for MD simulations; M.B.: formal analysis for microrheology experiments; C.R.: investigation and formal analysis for SAXS experiments; G.D.: conceptualization, methodology, investigation, and formal analysis for MD simulations; H.Z.: conceptualization, investigation, and formal analysis for mass spectrometry experiments; N.S.M.: conceptualization, investigation, and formal analysis for SAXS experiments; A.J.G.: conceptualization and funding acquisition for SAXS experiments; Z.S.: conceptualization, methodology, and funding acquisition for MPA and GUV experiments; B.S.S.: conceptualization, methodology, investigation, formal analysis, project administration, and funding acquisition.

## Competing interests

The authors declare no competing interests.
