## [Transparent Peer Review file · Nature Communications]

Phosphorylation Toggles the SARS-CoV-2 Nucleocapsid Protein Between Two Membrane-Associated Condensate States

Corresponding Author: Professor Benjamin Schuster

Version 0:

Reviewer comments:

Reviewer #1

(Remarks to the Author)

The work of Favetta et al. uses a comprehensive biophysical approach to evaluate the changes that phosphorylation induces on the material properties formed by the nucleocapsid protein (N) of SARS-CoV-2 and RNA. They show that these differences influence the condensate-membrane interaction that are mediated by different NTA anchored proteins: Nsp3 and M. They propose a model in which the protein phosphorylation state determines the condensate material properties and their interaction with different membranous organelles in the cell interior. Overall, the work is clear, well written, and the conclusions are supported by the experiments. I would recommend publication in Nature Communication after some concerns are addressed:

1) Page 8 line 177-178 “This suggests that N or pN and RNA condensates have no intrinsic ability to interact with lipid membranes of the composition tested”.

Here the authors should specify under the “tested conditions” including the buffer, for more accuracy. As the authors explain later in the text, the interaction between condensates and membranes is not only determined by the membrane composition, but also by the material properties of the condensates. In this sense, the material properties of the condensates can be greatly influenced by the buffer composition [1-3]. This means that even in the presence of certain ions the material properties of the condensates can change and therefore their interaction with membranes (most likely it would not be the same to use Tris+NaCl than PBS, for example). As the authors are testing only one specific membrane composition and buffer condition, I believe should be clarified and also discussed later in the text.

2) How is the efficiency of M and Nsp3 protein binding to the vesicles?

The authors cited reference 37 [4] that shows that when incorporating NTA lipids via electroformation using ITO glasses, the efficiency of NTA incorporation to the membrane is very low (compared to other methods); most likely because NTA lipids interact with the indium in the glass coating. This can lead to different percentages of NTA being incorporated on the membranes, and then different His-tag protein concentrations bound.

Authors should check the GFP intensity at the membrane after reconstitution with Nsp3 and M.

3) The interaction between the GUVs and the condensates will depend on the initial vesicle tension. If the vesicles are too tense, the interaction will be precluded. While membrane tension will be heterogeneous within a vesicle suspension, it is possible to reduce it via deflation, by increasing a few mM the external buffer solution. In the material and methods authors state that vesicles are electroformed using 300 mM sucrose, and the condensate buffer should have a higher osmolarity, at least 320 mOsM, is this right? Authors should be certain that the initial vesicle tension is not high, specially for the cases where no interaction/only binding is observed.

4) The middle panel in Figure 2E shows some clustering of the GFP signal. Is this membrane that is attaching to the condensates? The contrast of these images should be improved.

5) Is the data in Figures 2G-H coming from different samples?

6) In Figure 3B when M is put in contact with the N+RNA condensates, M appears to be binding only to the surface of the condensates, is this correct? or is it inducing the formation of hollow condensates? Could you provide the bright-field images?

7) While sections three and four are relevant to the work, I believe authors should credit the previous work that has been done in this regard. This would not reduce the “novelty”, but I think that for example the beginning of section 3 is unfair: Page 13, line 315: “Based on our membrane binding experiments, we hypothesized that phosphorylation makes N condensates more fluid...”

Authors state in the introduction that it was already shown that the material properties of N condensates changed with the phosphorylation, and that they hypothesized that this could be the basis of the dual role of N during the viral lifecycle [5] (the preprint should be change by the published citation). While the authors do a more complete analysis on the material properties of the condensates, and use particular RNA sequences, the manner in which this section is presented should contemplate the previous findings. For example, the FRAP results showing that N is less mobile than pN, matches the findings of Carlson et al., and this is not mentioned.

Page 14 line 329-330 reads: “Overall, our FRAP results provide insight towards understanding the membrane interactions observed in previous experiments”.

I believed it was the other way around: according to the introduction the authors started with the hypothesis that different material properties should imply different membrane interactions.

8) The same for section 4. The effect of RNA on the material properties of condensates has been described. See for example the papers from the Priya Banerjee group: Figure 6 in [6] and also [7]. These papers are relevant to the discussion, specially since authors state that “These findings can be applied to better understand other protein-RNA condensates as well as to design more comprehensive studies on the effect of RNA on condensates” (Page 25 lines 597-598). I believe the works from Banerjee’s lab are quite comprehensive.

9) Finally, in the discussion there is (from my point of view) an unnecessary clarification saying that in vitro systems are often simpler, and that in this work the reconstitution is more “physiological”. I strongly disagree with this claim. This work is done with reconstituted protein fragments, synthetic lipids (NTA lipids do not even exist in nature) and RNA fragments, then, in which sense it is more physiological? Again, considering that only one buffer condition and membrane composition is tested, I would be more careful with this type of statements. More likely in the crowded environment of the cell the condensate and membrane compositions and interactions will be different to what it is measured in vitro. Having said this, in vitro systems are valuable tools that have allowed us to measure and understand most of the biology that we know today; we only need to be aware of their limitations and not to push our conclusions far beyond of what is being measured.

A minor comment: the first paragraph explaining the condensate formation on the microscopy section in material and methods, should be revised for clarity.

References:

- [1] L.M. Jawerth, M. Ijavi, M. Ruer, S. Saha, M. Jahnel, A.A. Hyman, F. Jülicher, E. Fischer-Friedrich, Salt-Dependent Rheology and Surface Tension of Protein Condensates Using Optical Traps, *Physical Review Letters*, 121 (2018) 258101.
- [2] A. Mangiarotti, M. Siri, N.W. Tam, Z. Zhao, L. Malacrida, R. Dimova, Biomolecular condensates modulate membrane lipid packing and hydration, *Nature Communications*, 14 (2023) 6081.
- [3] L. Zhu, Y. Pan, Z. Hua, Y. Liu, X. Zhang, Ionic Effect on the Microenvironment of Biomolecular Condensates, *Journal of the American Chemical Society*, 146 (2024) 14307-14317.
- [4] S. Pramanik, J. Steinkühler, R. Dimova, J. Spatz, R. Lipowsky, Binding of His-tagged fluorophores to lipid bilayers of giant vesicles, *Soft Matter*, 18 (2022) 6372-6383.
- [5] C.R. Carlson, J.B. Asfaha, C.M. Ghent, C.J. Howard, N. Hartooni, M. Safari, A.D. Frankel, D.O. Morgan, Phosphoregulation of Phase Separation by the SARS-CoV-2 N Protein Suggests a Biophysical Basis for its Dual Functions, *Molecular Cell*, 80 (2020) 1092-1103.e1094.
- [6] I. Alshareedah, M.M. Moosa, M. Pham, D.A. Potoyan, P.R. Banerjee, Programmable viscoelasticity in protein-RNA condensates with disordered sticker-spacer polypeptides, *Nature Communications*, 12 (2021) 6620.
- [7] R. Laghmach, I. Alshareedah, M. Pham, M. Raju, P.R. Banerjee, D.A. Potoyan, RNA chain length and stoichiometry govern surface tension and stability of protein-RNA condensates, *iScience*, 25 (2022) 104105.

Reviewer #2

(Remarks to the Author)

This manuscript entitled "Phosphorylation Toggles the SARS-CoV-2 Nucleocapsid Protein Between Two Membrane-Associated Condensate States" reveals an important regulatory role of phosphorylation in modulating the phase separation behavior, membrane interactions, and functional states of the SARS-CoV-2 nucleocapsid (N) protein. The work addresses a significant gap in understanding the mechanistic basis of N's dual roles in RNA replication and viral genome packaging. By integrating multidisciplinary approaches, including fluorescence polarization, FRAP, SAXS, and in vitro reconstitution of

membrane interactions, the authors provide evidence supporting their hypothesis that phosphorylation acts as a molecular switch. The findings are quite interesting, offering novel insights into the functions of SARS-CoV-2 N protein. Prior to any final recommendation, several aspects of the manuscript need to be further improved.

Major points

1. While the authors present several experimental evidence demonstrating the role of phosphorylation in regulating the distinct functions of the N protein, all these experiments were conducted in vitro. To strengthen the physiological relevance of their findings, the authors should verify their conclusions using at least cell-based models. For example, the authors propose that phosphorylated N mediates viral genome replication, while unmodified N facilitates viral assembly. However, the detection of the phosphorylated N and unmodified N proteins during the viral infection process should be performed. This would give clear evidence to support the phosphorylation event in cells upon viral infections.
2. In Figure 1B, the authors proposed multiple phosphorylation sites on the N protein. A LC-MS/MS analysis would be necessary to delineate the exact phosphorylation sites of N.
3. Based on the identification of the phosphorylation sites, the transcription and replication-competent SARS-CoV-2 virus-like-particles (trVLPs) (eg, PMID: 33711082) should be used to verify the role of N phosphorylation by expressing different N phosphorylation site mutants.
4. The authors propose targeting relevant kinases (e.g., GSK-3 β and SRPK1) as novel COVID therapeutics. More compelling data could be acquired using the kinase-deficient cell models to study the SARS-CoV-2 infections.

Minor

1. In Fig.2D, the authors compared the behavior of N and pN, why data from different timepoint settings are shown.
2. N is known to regulate the innate immune response, does phosphorylated N play a role in this?
3. Several references are missing (PMIDs: 32901111, 33837182, 34239064).

Reviewer #3

(Remarks to the Author)

The paper by Favetta et al. focuses on the role of phosphorylation in the N protein of CoV-2. I was specifically asked to focus on the SAXS part of the paper. However, I cannot help but notice that when N phosphorylation is discussed in the introduction, the interaction with 14-3-3 proteins that is phosphorylation dependent (14-3-3 make up 1% of all soluble proteins in a cell) is neither taken into account nor discussed. I consider this a significant drawback. Structures of N in complex with 14-3-3 exist and are well described in the literature and certainly N behaves differently when bound to 14-3-3. Concerning SAXS analysis, the Guinier plot (Figure S11) of N at 4 mg/ml reveals a little bit of aggregation, these data should be used with caution, or preferable not used at all. What would really help the SAXS analysis is to calculate theoretical scattering curves based on computer simulations or the phosphorylated N and non-phosphorylated N (for non-phosphorylated N this was already done). However, this would be a significant effort and while improving a lot Figure 5 section F-J it would not change the main message of the study, so my recommendation to the editors is not to insist on it. I also have few comments beyond SAXS.

Major comments:

- 1) "GUVs were made with a lipid composition meant to approximate the human ER membrane, with 60% DOPC, 25% DOPE, 10% DOPS, and the addition of 5% Ni-NTA lipids."

Seriously, a membrane without cholesterol was prepared to mimic ER membrane?

- 2) We performed experiments at two protein concentrations: low concentration, at which N is in its monomeric form, and high concentration, at which dimers form if the dimerization domain is present.

I believe that the experiments were done correctly. However, a little more explanation of how single molecule FRET was done in high concentration would help. Was the labeled protein mixed with unlabeled protein or some other "trick" was used?

Minor comments:

- 1) "The N-terminal folded domain (NTD) strongly binds with specific viral RNA elements, stabilizing the RNP complex"

RNP should be defined when it is used for the first time.

- 2) "The central disordered region acts as a linker that allows for conformational flexibility. More specifically, the serine/arginine (SR)-rich region (aa 175 – 206; Figure 1B) within the linker is known to participate in both protein–protein and protein–RNA interactions."

SAXS derived structures of N are available in the literature, they should be discussed here.

Version 1:

Reviewer comments:

Reviewer #1

(Remarks to the Author)

The authors have responded to my comments and suggestions, and I have no further observations. The work has improved and gained strength, especially with the incorporation of the molecular dynamics simulation results. Therefore, I recommend publication in Nature Communications.

Reviewer #2

(Remarks to the Author)

The authors addressed my concerns with discussions and new data.

Reviewer #3

(Remarks to the Author)

All my concerns were properly addressed and I am now happy to fully recommend the manuscript by Favetta et al. for publication.

Authors' Response to Reviewers' Comments

We thank the editor and three reviewers for their valuable feedback on our manuscript. The reviews highlight that our work represents a “comprehensive biophysical approach to evaluate the changes that phosphorylation induces” on N protein and that it “reveals an important regulatory role of phosphorylation”. However, in summarizing the reviewers' feedback, the editor highlighted areas for improvement. In particular, the reviewers had questions about the design of our experiments with Giant Unilamellar Vesicles, contextualizing our findings with previous literature, validating our conclusions in cell-based models, verifying phosphorylation of N protein using additional mass spectrometry, and validating our SAXS results using coarse grained modeling of N.

Here and in the revised manuscript, we address the reviewers' major and minor comments about the experiments and their interpretation. We have done so by incorporating data from new experiments and modifying the text (major text changes are highlighted in the manuscript). In particular, we have:

- Repeated our GUV experiments with more complex membrane compositions, including a range of cholesterol concentrations, that more closely resemble the composition of the endoplasmic reticulum membrane and viral capsid membrane.
- Verified that binding of membrane protein fragments GFP-M¹⁰⁴⁻²²² and GFP-Nsp3¹⁻¹¹¹ to GUVs is similar and that the osmolarity of buffer compositions is as expected.
- Contextualized our FRAP, RNA, and SAXS experiments to refer to previously published data, highlighting our contributions more specifically.
- Included a section in the discussion on the novel contributions our work. We highlight that our experiments were motivated by findings from cell-based experiments for which no biophysical mechanism was understood. For example, N's interactions with membranes were known to change from the early stages of viral replication to later stages during viral assembly, but the biophysical mechanism determining these interactions was not known.
- Including a discussion of the limitations of our in vitro methods, particularly that we have simplified the system to study a subset of important proteins and environmental conditions.
- Completed an in-depth mass spectrometry analysis of the phosphorylation profile of N protein to confirm the expected phosphorylation sites.
- Generated an ensemble from a coarse-grained simulation of the unmodified and phosphomimetic N protein and compared the results with our SAXS data to deepen our insights into the molecular conformation change that occurs upon phosphorylation.

We believe that these revisions have significantly improved our manuscript. Our work represents an impactful contribution, highlighting how a suite of biophysical characterization techniques can be used together to gain an in-depth understanding of the structure-behavior relationship for N protein. Our work is quite timely as well, building on recently published literature on the role of phase separation in the SARS-CoV-2 lifecycle, providing complementary evidence to several recent contributions from the NMR field on N protein structure, and

highlighting the importance of recent work understanding the factors determining condensate-membrane interactions. Thank you for considering our work for publication in *Nature Communications*.

Reviewer #1 (Remarks to the Author):

The work of Favetta et al. uses a comprehensive biophysical approach to evaluate the changes that phosphorylation induces on the material properties formed by the nucleocapsid protein (N) of SARS-CoV-2 and RNA. They show that these differences influence the condensate-membrane interaction that are mediated by different NTA anchored proteins: Nsp3 and M. They propose a model in which the protein phosphorylation state determines the condensate material properties and their interaction with different membranous organelles in the cell interior. Overall, the work is clear, well written, and the conclusions are supported by the experiments. I would recommend publication in Nature Communication after some concerns are addressed:

We thank the reviewer for their valuable feedback and for highlighting the strengths of our work.

1) Page 8 line 177-178 “This suggests that N or pN and RNA condensates have no intrinsic ability to interact with lipid membranes of the composition tested”. Here the authors should specify under the “tested conditions” including the buffer, for more accuracy. As the authors explain later in the text, the interaction between condensates and membranes is not only determined by the membrane composition, but also by the material properties of the condensates. In this sense, the material properties of the condensates can be greatly influenced by the buffer composition [1-3]. This means that even in the presence of certain ions the material properties of the condensates can change and therefore their interaction with membranes (most likely it would not be the same to use Tris+NaCl than PBS, for example). As the authors are testing only one specific membrane composition and buffer condition, I believe should be clarified and also discussed later in the text.

We apologize that our explanation was lacking in detail. We edited the text in page 9, lines 194-195, to include the lipid composition and buffer conditions used. In addition, we included a paragraph in the discussion section that speaks to the limitations of the study. There, we noted that the crowded environment of the cell may affect the behavior of N, both in terms of its material properties and membrane interactions.

We would also like to note that given the comments from reviewer 2, we repeated our GUV experiments with a second set of membrane compositions that include a range of cholesterol concentrations. This data can be found in Supplementary Figure 3, and copied into page 23 of this response letter, and described in a new paragraph in the Results (page 10, lines 219-235).

2) How is the efficiency of M and Nsp3 protein binding to the vesicles? The authors cited reference 37 [4] that shows that when incorporating NTA lipids via electroformation using ITO glasses, the efficiency of NTA incorporation to the membrane is very low (compared to other methods); most likely because NTA lipids interact with the indium in the glass coating. This can lead to different percentages of NTA being incorporated on the membranes, and then different His-tag protein concentrations bound. Authors should check the GFP intensity at the membrane after reconstitution with Nsp3 and M.

Thank you for raising this question. We verified that GFP-M¹⁰⁴⁻²²² and GFP-Nsp3¹⁻¹¹¹ have similar binding affinities to GUVs with NTA(Ni) lipids. We measured the fluorescence intensity at the GUV surface across 6 batches of GUVs prepared using confocal imaging. We observe some variability in binding strength across batches and between each protein, but the differences were not significant. The maximum intensity for GFP-M¹⁰⁴⁻²²² was 0.84 ± 0.12 and for GFP-Nsp3¹⁻¹¹¹, 0.77 ± 0.13 . The difference in maximum fluorescence intensity was not statistically significant ($p = 0.2$ using a two-sided student's t-test). This data was included as Supplementary Figure 4 and is copied below.

Figure S4. Degree of tethering of membrane protein fragments to the GUV surface. A) (top) Representative confocal images from three GUV batches with 3 μM GFP-M¹⁰⁴⁻²²² added to the sample. Fluorescence intensities were quantified as line profiles across individual GUVs as indicated by the example dashed white line. Scale bar = 5 μm. (bottom) Fluorescence intensity profiles across one GUV from each batch made ($n = 6$), normalized by GUV size. Intensities were also divided by 2^{16} , the dynamic range of the images, such that intensities scale from 0 to 1. B) (top) Representative confocal images from three GUV batches with 3 μM GFP-Nsp3¹⁻¹¹¹ added to the sample. Scale bar = 5 μm. (bottom) Fluorescence intensity profiles across one GUV from each batch made ($n = 6$), normalized by GUV size. Intensities were also divided by 216, the dynamic range of the images, such that intensities scale from 0 to 1. C) Maximum intensity measured in the GFP-M¹⁰⁴⁻²²² and GFP-Nsp3¹⁻¹¹¹ samples. The difference in intensity is not statistically significant ($p = 0.2$). p value was determined using a two-sided student's t-test.

3) The interaction between the GUVs and the condensates will depend on the initial vesicle tension. If the vesicles are too tense, the interaction will be precluded. While membrane tension

will be heterogeneous within a vesicle suspension, it is possible to reduce it via deflation, by increasing a few mM the external buffer solution. In the material and methods authors state that vesicles are electroformed using 300 mM sucrose, and the condensate buffer should have a higher osmolarity, at least 320 mOsM, is this right? Authors should be certain that the initial vesicle tension is not high, specially for the cases were no interaction/only binding is observed.

Thank you for the suggestion. We verified the osmolality of our buffers: the osmolality of the sucrose solution (301 ± 1 mOsM) was lower than that of the condensate buffer (324 ± 1 mOsM), as expected. This data represents the average of five independent buffer preparations. This should have ensured the vesicles were not too tense during experiments. This data was added to the methods section.

4) The middle panel in Figure 2E shows some clustering of the GFP signal. Is this membrane that is attaching to the condensates? The contrast of these images should be improved.

Thank you for the suggestions. Yes, we believe that clustering of the GFP signal in Figure 2E is due to condensate binding to membranes driven both by their inherent binding affinity and the force of attraction of the optical tweezers, which are used at low power to hold condensates in the imaging plane over the acquisition of the timelapse. We improved the contrast of images in 2E and increased their size to facilitate viewing.

5) Is the data in Figures 2G-H coming from different samples?

We apologize the figure caption was not sufficiently detailed. Data in Figures 2G and 2H were calculated from the same image data set, by re-doing angle calculations. The image data set represented three independent samples for each protein + GUV combination. We have now included this information in the image caption.

6) In Figure 3B when M is put in contact with the N+RNA condensates, M appears to be binding only to the surface of the condensates, is this correct? or is it inducing the formation of hollow condensates? Could you provide the bright-field images?

We thank the reviewer for the suggestion and apologize the figure was not clear. We observe M binding only to the surface of condensates and do not observe formation of hollow condensates upon addition of the membrane protein fragments. We have now included brightfield images in the revised Figure 3 (copied below).

Figure 3: N and pN interaction with membrane proteins, M and Nsp3. A) Schematic of the experimental setup where 6xHis-GFP-M¹⁰⁴⁻²²² or 6xHis-GFP-Nsp3¹⁻¹¹¹ is added to a condensate sample and partitioning of GFP-tagged protein is quantified using confocal microscopy. B) Representative confocal images from condensates composed of 40 μM N or pN plus either no RNA or 300 nM RNA; plus either 2 μM 6xHis-GFP-M¹⁰⁴⁻²²² or 6xHis-GFP-Nsp3¹⁻¹¹¹. 6xHis-GFP-M¹⁰⁴⁻²²² binds to N but not pN condensates while 6xHis-GFP-Nsp3¹⁻¹¹¹ partitions in regardless of choice of N vs. pN. Normalized line profiles across condensates, representing averages of at least n = 20 condensates from 3 independent trials. Scale bars = 5 μm. C) Quantification of partitioning of GFP-tagged proteins into N or pN condensates by dividing average fluorescence inside condensates (if any) by the average background fluorescence. Error bars represent one standard deviation (±1 s.d.). p values were determined using two-way ANOVA followed by post hoc Tukey's test. *p < 0.01.

7) While sections three and four are relevant to the work, I believe authors should credit the previous work that has been done in this regard. This would not reduce the “novelty”, but I think

that for example the beginning of section 3 is unfair: Page 13, line 315: “Based on our membrane binding experiments, we hypothesized that phosphorylation makes N condensates more fluid...” Authors state in the introduction that it was already shown that the material properties of N condensates changed with the phosphorylation, and that they hypothesized that this could be the basis of the dual role of N during the viral lifecycle [5] (the preprint should be change by the published citation). While the authors do a more complete analysis on the material properties of the condensates, and use particular RNA sequences, the manner in which this section is presented should contemplate the previous findings. For example, the FRAP results showing that N is less mobile than pN, matches the findings of Carlson et al., and this is not mentioned. Page 14 line 329-330 reads: “Overall, our FRAP results provide insight towards understanding the membrane interactions observed in previous experiments”. I believed it was the other way around: according to the introduction the authors started with the hypothesis that different material properties should imply different membrane interactions.

We apologize that our text did not properly refer to the previous literature. We edited the material properties section (page 17, lines 338-339) to make clearer how our findings agree with previously published FRAP data. We reframed our manuscript to state that the novelty in our experiment is in measuring how the material properties affect membrane interactions, rather than simply measuring the material properties themselves (page 17, lines 329-331 and page 18, lines 347-348). We would also like to note that Carlson et al. only completed FRAP experiments with a phosphomimetic version of N, not in vitro phosphorylated N. Although the trends match, we observe more recovery with pN than Carlson observed with phosphomimetic N.

8) The same for section 4. The effect of RNA on the material properties of condensates has been described. See for example the papers from the Priya Banerjee group: Figure 6 in [6] and also [7]. These papers are relevant to the discussion, specially since authors state that “These findings can be applied to better understand other protein-RNA condensates as well as to design more comprehensive studies on the effect of RNA on condensates” (Page 25 lines 597-598). I believe the works from Banerjee’s lab are quite comprehensive.

We thank the reviewer for the suggestion. We now refer to and cite previous work investigating the effect of RNA structure on condensate material properties in page 21, lines 420-422. We agree that the Banerjee group has previously shown that condensate material properties depend on RNA structure [1]. However, we also highlight the difference between our works: the Banerjee group took a reductionist approach with a fully disordered protein and a short structured RNA, while we focus on a naturally occurring partially folded protein and longer, more structurally complex, virus-derived RNA sequences.

[1] I. Alshareedah, M.M. Moosa, M. Pham, D.A. Potoyan, P.R. Banerjee, Programmable viscoelasticity in protein-RNA condensates with disordered sticker-spacer polypeptides, Nature Communications, 12 (2021) 6620.

9) Finally, in the discussion there is (from my point of view) an unnecessary clarification saying that in vitro systems are often simpler, and that in this work the reconstitution is more “physiological”. I strongly disagree with this claim. This work is done with reconstituted protein

fragments, synthetic lipids (NTA lipids do not even exist in nature) and RNA fragments, then, in which sense it is more physiological? Again, considering that only one buffer condition and membrane composition is tested, I would be more careful with this type of statements. More likely in the crowded environment of the cell the condensate and membrane compositions and interactions will be different to what it is measured in vitro. Having said this, in vitro systems are valuable tools that have allowed us to measure and understand most of the biology that we know today; we only need to be aware of their limitations and not to push our conclusions far beyond of what is being measured.

We agree with your insightful feedback and appreciate the in-depth assessment of the importance and limitations of in vitro work. We revised this paragraph (page 34, lines 709-727) to reflect a discussion on the advantages and limitations of our in vitro study. In this revised version, we note that in infected cells there are many factors that may influence the behavior of N condensates, but that turning to in vitro methods allowed us to study the biophysical mechanisms behind the complex behaviors observed in cells.

A minor comment: the first paragraph explaining the condensate formation on the microscopy section in material and methods, should be revised for clarity.

We apologize the text was not clear. We have revised it accordingly.

References:

- [1] L.M. Jawerth, M. Ijavi, M. Ruer, S. Saha, M. Jahnel, A.A. Hyman, F. Jülicher, E. Fischer-Friedrich, Salt-Dependent Rheology and Surface Tension of Protein Condensates Using Optical Traps, *Physical Review Letters*, 121 (2018) 258101.
- [2] A. Mangiarotti, M. Siri, N.W. Tam, Z. Zhao, L. Malacrida, R. Dimova, Biomolecular condensates modulate membrane lipid packing and hydration, *Nature Communications*, 14 (2023) 6081.
- [3] L. Zhu, Y. Pan, Z. Hua, Y. Liu, X. Zhang, Ionic Effect on the Microenvironment of Biomolecular Condensates, *Journal of the American Chemical Society*, 146 (2024) 14307-14317.
- [4] S. Pramanik, J. Steinkühler, R. Dimova, J. Spatz, R. Lipowsky, Binding of His-tagged fluorophores to lipid bilayers of giant vesicles, *Soft Matter*, 18 (2022) 6372-6383.
- [5] C.R. Carlson, J.B. Asfaha, C.M. Ghent, C.J. Howard, N. Hartooni, M. Safari, A.D. Frankel, D.O. Morgan, Phosphoregulation of Phase Separation by the SARS-CoV-2 N Protein Suggests a Biophysical Basis for its Dual Functions, *Molecular Cell*, 80 (2020) 1092-1103.e1094.
- [6] I. Alshareedah, M.M. Moosa, M. Pham, D.A. Potoyan, P.R. Banerjee, Programmable viscoelasticity in protein-RNA condensates with disordered sticker-spacer polypeptides, *Nature Communications*, 12 (2021) 6620.

[7] R. Laghmach, I. Alshareedah, M. Pham, M. Raju, P.R. Banerjee, D.A. Potoyan, RNA chain length and stoichiometry govern surface tension and stability of protein-RNA condensates, *iScience*, 25 (2022) 104105.

Reviewer #2 (Remarks to the Author):

This manuscript entitled "Phosphorylation Toggles the SARS-CoV-2 Nucleocapsid Protein Between Two Membrane-Associated Condensate States" reveals an important regulatory role of phosphorylation in modulating the phase separation behavior, membrane interactions, and functional states of the SARS-CoV-2 nucleocapsid (N) protein. The work addresses a significant gap in understanding the mechanistic basis of N's dual roles in RNA replication and viral genome packaging. By integrating multidisciplinary approaches, including fluorescence polarization, FRAP, SAXS, and in vitro reconstitution of membrane interactions, the authors provide evidence supporting their hypothesis that phosphorylation acts as a molecular switch. The findings are quite interesting, offering novel insights into the functions of SARS-CoV-2 N protein. Prior to any final recommendation, several aspects of the manuscript need to be further improved.

We thank the reviewer for noting that our work is quite interesting and provides novel insights into the functions of SARS-CoV-2 N protein.

1. While the authors present several experimental evidence demonstrating the role of phosphorylation in regulating the distinct functions of the N protein, all these experiments were conducted in vitro. To strengthen the physiological relevance of their findings, the authors should verify their conclusions using at least cell-based models. For example, the authors propose that phosphorylated N mediates viral genome replication, while unmodified N facilitates viral assembly. However, the detection of the phosphorylated N and unmodified N proteins during the viral infection process should be performed. This would give clear evidence to support the phosphorylation event in cells upon viral infections.

We thank the reviewer for the suggestion. Our goal in turning to in vitro experiments was to provide quantitative biophysical explanations to explain several observations regarding N protein behavior that were reported by others investigating SARS-CoV-2 infected cells or N expressed in mammalian cell culture. Given the limitations of experiments in cells, the mechanisms that resulted in the observed behaviors were unknown.

For example, you noted above that the detection of phosphorylated N and unmodified N proteins during the viral infection process should be performed. Several groups have detected phosphorylated N inside cells and unmodified N in virions using mass spectrometry [1-3]. Yaron et al. reported 39 phosphorylation sites on N protein purified from infected cells, 18 of which are in the protein's SR-rich region. Ino et al. found N in virions to only have one consistently phosphorylated site (Ser79), located outside of the SR-rich region. Our aim was therefore not to verify that N is phosphorylated in cells while unmodified in virions. Using this previously published data as a starting point, we asked the question: Why is N found in different phosphorylation states depending on the stage of the viral lifecycle? How does phosphorylation

of N in infected cells affect its structure and therefore the function of the protein? In our manuscript, we detail at a molecular level how phosphorylation changes N protein's conformation, which affects its ability to bind to RNA and to other viral membrane proteins, consequently changing how the protein interacts with membranes.

1. Patton, D., Stohlman, S. A., Fleming, J. & Lai, A. M. C. Synthesis and Subcellular Localization of the Murine Coronavirus Nucleocapsid Protein. *Virology* 130, 527–532 (1983).
2. Yaron, T. M. et al. Host protein kinases required for SARS-CoV-2 nucleocapsid phosphorylation and viral replication. *Sci. Signal.* 15, 1–16 (2022).
3. Ino, Y. et al. Phosphopeptide enrichment using Phos-tag technology reveals functional phosphorylation of the nucleocapsid protein of SARS-CoV-2. *J. Proteomics* 255, 104501 (2022).

A second observation by others investigating N in infected cells is that during viral replication, when N is phosphorylated, it forms a thin layer around viral replication organelles (vROs), double membrane vesicles filled with viral RNA commonly found in infected cells [4]. However, during virion assembly, complexes of unmodified N and RNA remain as small, spherical structures while the viral capsid is engulfed by the ER-Golgi intermediate complex (ERGIC) membrane [5]. Again, our goal was not to verify these different behaviors of N inside cells. We used previously published evidence from these cell-based experiments to hypothesize that N protein's interactions with membranes is affected by phosphorylation. In these cell-based models, researchers were not able to investigate a biophysical mechanism that would explain the observed behaviors. In our manuscript, we are able to modulate the material properties of N condensates and the types of viral membrane proteins present to understand which factors affect protein-membrane interactions.

4. Scherer, K. M. et al. SARS-CoV-2 nucleocapsid protein adheres to replication organelles before viral assembly at the Golgi/ERGIC and lysosome-mediated egress. *Sci. Adv.* 8, (2022).
5. Wolff, G. et al. A molecular pore spans the double membrane of the coronavirus replication organelle. *Science* (80-.). 369, 1395–1398 (2020).

In addition, researchers have also shown that the dynamic movement of N differs when expressed in mammalian cells compared to when expressed in bacterial culture and purified for in vitro experiments. Using FRAP, it has been shown that N in cells shows up to 80% recovery in ~1 minute, while in vitro, less recovery is observed (20 – 40%) in 2 – 4 minutes [6-7]. These findings suggest that modification of N in cells changes the behavior of the protein. Previously, an in vitro study used FRAP to compare N and a phosphomimetic version of N in vitro and found that the phosphomimetic version appeared more dynamic [8]. From the starting point of these observations, we (1) further quantified the material properties of N and pN, and (2) investigated the mechanism through which phosphorylation affects the material properties of N.

6. Lu, S. et al. The SARS-CoV-2 nucleocapsid phosphoprotein forms mutually exclusive condensates with RNA and the membrane-associated M protein. *Nat. Commun.* 12, (2021).

7. Zhao, D. et al. Understanding the phase separation characteristics of nucleocapsid protein provides a new therapeutic opportunity against SARS-CoV-2. *Protein & Cell*. 12, (2021)
8. Carlson, C. R. et al. Phosphoregulation of Phase Separation by the SARS-CoV-2 N Protein Suggests a Biophysical Basis for its Dual Functions. *Mol. Cell* 80, 1092-1103.e4 (2020).

Therefore, we believe that the experiments you suggested we should conduct in cells to strengthen our results have already been completed and were instead the inspiration for our in vitro studies. Our aim was to explain the biophysical mechanisms determining the protein's behavior observed in cells.

We would also like to note that we agree that our in vitro experiments have limitations due to the simplification of the system. We have now included a paragraph in the discussion section (page 34, lines 709-727) that speaks to the advantages and limitations of our approach. In this section, we suggest further experiments that a laboratory with a BSL-3 setup can do, such as treating infected cells with kinase inhibitors and investigating the localization of N by fixing and staining for the appropriate proteins (the effects of targeting infected cells with kinase inhibitors have been investigated, but the effects on N's membrane interaction have not been reported).

2. In Figure 1B, the authors proposed multiple phosphorylation sites on the N protein. A LC-MS/MS analysis would be necessary to delineate the exact phosphorylation sites of N.

We thank the reviewer for the suggestion. First, we would like to highlight that the phosphorylation sites on N protein in infected cells have been previously characterized by several groups [1-5]. To verify that our in vitro phosphorylated N had the same pattern of phosphorylation as that found in cells, we have conducted two additional mass spectrometry experiments:

- By digesting N and pN protein with pepsin protease and conducting LC/MS on the fragments, we identified a peptide that had up to 8 phosphate groups included. The fragment comprised residues 172 to 219, which encompasses the protein's SR-rich region. Phosphorylation was not identified elsewhere in the protein.
- By digesting N and pN protein with trypsin protease and conducting LC/MS on the peptide fragments, we were able to specifically identify 6 sites that were phosphorylated (S176, S180, S184, S184, T198, S202, S206). These had all been previously shown to be targets of the SRPK1 and GSK3 β kinases. Protease digestion patterns did not allow us to specifically identify other sites.

We included these results in Supplementary Figure 1 B and C which are copied below. Thus, our mass spectrometry results agree with the SARS-CoV-2 literature and confirms that our in vitro phosphorylation protocol successfully mimicked phosphorylation of the SR region in infected cells.

A. Unmodified N, theoretical mass: 47300 Da

Phosphorylated N, theoretical mass with 9 phosphate groups: 48020 Da

B.

Peptide: 172 AEGSRGGSQASSRSSSRSRNSSRNSTPGSSRGTS~~PAR~~MAGNGGDAAL
219

C. Peptide: 170 GFYAEGSR 177

Peptide: 170 GFYAEGSRGG**S**QASSR 185

Peptide: 170 GFYAEGSRGG**S**QASSR 185

Peptide: 196 NSTPG**S**RGTS**S**PAR 209

Peptide: 196 NST**TPGSS**SRGT**SPAR** 209

D. Phos-tag gel electrophoresis

1. N
2. N + SRPK + GSK3 β + Rxn buffer + 120 min
3. Empty
4. Rxn mixture at 0 min
5. Rxn mixture at 15 min
6. Rxn mixture at 30 min
7. Rxn mixture at 45 min
8. Rxn mixture at 60 min
9. Rxn mixture at 75 min
10. Rxn mixture at 90 min
11. Rxn mixture at 105 min
12. Rxn mixture at 120 min

Figure S1. Confirmation of N protein phosphorylation. A) Mass spectra for N protein prior to phosphorylation and following phosphorylation, showing a mass increase of the main peak from 47.3 kDa to 48.0 kDa. This mass increase represents the addition of 9 (+ 720 Da) phosphate groups. B) Following digestion of N / pN protein with pepsin protease, we conducted LC-MS on the peptide fragments. (Top) Mass spectra of peptides from a control sample (N + SRPK1 + GSK3 β without ATP). A peptide containing the SR-rich region (residues 172-219) was identified and labelled as “unmodified” (with sequence noted above). (Bottom) Mass spectra of peptides from a sample with phosphorylated N (N + SRPK1 + GSK3 β + ATP). The peak representing the unmodified peptide can no longer be identified. New peaks with the expected m/z of peptides with 3 – 8 phosphate groups added to the peptide can be identified and are labelled. The presence of peptides with a range of phosphorylation states indicates variability in the number of sites that are successfully phosphorylated per protein. C) Representative mass spectra identifying phosphorylation sites in peptides from pN protein. Following digestion of pN with trypsin protease, we conducted LC-MS on the peptide fragments. 5 peptides were identified from which phosphorylation sites could be precisely identified: S176, S180, S184, S184, T198, S202, S206, which are highlighted in bold above the respective spectra. Additional phosphorylation sites could be detected in other peptides, but the precise site of phosphorylation could not be identified due to the protease digestion patterns. All sites had been previously identified by Yaron et al.,

2022. D) SuperSep Phos-tag SDS-PAGE of N protein sample before, after, and during phosphorylation reaction at the timepoints indicated.

1. Yaron, T. M. et al. Host protein kinases required for SARS-CoV-2 nucleocapsid phosphorylation and viral replication. *Sci. Signal.* 15, 1–16 (2022).
2. Davidson AD et al. Characterization of the transcriptome and proteome of SARS-CoV-2 reveals a cell passage induced in-frame deletion of the furin-like cleavage site from the spike glycoprotein. *Genome Med* 12, 68 (2020).
3. Bouhaddou M et al. , The Global Phosphorylation Landscape of SARS-CoV-2 Infection. *Cell* 182, 685–712 e619 (2020).
4. Hekman RM et al. , Actionable Cytopathogenic Host Responses of Human Alveolar Type 2 Cells to SARS-CoV-2. *Molecular Cell*, (2020).
5. Klann K et al. , Growth factor receptor signaling inhibition prevents SARSCoV-2 replication. *Molecular Cell*, (2020).

3. Based on the identification of the phosphorylation sites, the transcription and replication-competent SARS-CoV-2 virus-like-particles (trVLPs) (eg, PMID: 33711082) should be used to verify the role of N phosphorylation by expressing different N phosphorylation site mutants.

We thank the reviewer for highlighting this literature. As the phosphorylation sites of N protein had already been previously mapped, several groups have already verified the effect of phosphorylation-null mutants of N on the production of virus-like particles.

- In Ju et al., several key phosphorylation target residues were mutated (S176, S413, S176/413, S105, S183, S188, S206, S188/206). Note that S188 and S206 are the two sites that SRPK1/2 phosphorylates which then primes the SR-rich region for phosphorylation in up to 8 more sites by GSK3 α/β . The researchers found “most of the phosphorylation null mutants were able to assemble virus-like particles with comparable or slightly reduced efficiencies than WT. However, S188A/S206A double mutations completely abolished N function.” Here, N function was measured as the ability to reinfect cells. These results showed that N can still perform its functions if partially phosphorylated but will not be able to support viral replication without any phosphorylation.
- In Syed et al., researchers focused on mutations to T198, S202 and S206 (where S206 is one of two priming sites for SRPK1/2). They found that individual substitutions to alanine on those sites, such that N was only partially phosphorylated, resulted in increased VLP production. These results suggest that reducing phosphorylation of N makes it better suited for its function in new virion assembly. Researchers then studied the effects of S188A/S206A with single cycle infectious particles. They found that this double mutant, which cannot be phosphorylated, showed a 5x reduction in post-entry replication within cells. They conclude that “phosphorylation of the SR motif inhibits assembly but improves genome replication, suggesting an evolutionary trade-off between these functions”.

Together, these findings show that phosphorylation can enhance the ability of N to conduct the initial steps of viral infection but inhibit the final stages of viral assembly. Our in vitro results suggest a mechanism that explains these findings, wherein phosphorylation reduces the binding affinity between N and RNA, as well as N and M protein. This allows pN to conduct its dynamic roles in the initial steps of replication, while those interactions are needed by N in the final stages of replication for viral assembly. In other words, we conducted biophysical measurements that uncovered the mechanisms through which N can behave in the ways observed in cells.

Please note that we have now included these references in our introduction section (page 3, lines 62-64) to strengthen our argument that regulation of phosphorylation of N may act as a timer in the viral lifecycle.

1. Ju X et al. A novel cell culture system modeling the SARS-CoV-2 life cycle. *PLoS Pathog.* 2021 Mar 12;17(3)
2. Syed AM et al. SARS-CoV-2 evolution balances conflicting roles of N protein phosphorylation. *PLoS Pathog.* 2024 Nov 21;20(11)

4. The authors propose targeting relevant kinases (e.g., GSK-3 β and SRPK1) as novel COVID therapeutics. More compelling data could be acquired using the kinase-deficient cell models to study the SARS-CoV-2 infections.

We thank the reviewer for the suggestion. As we had noted in our discussion, others have already investigated the targeting of kinases in SARS-CoV-2 infection models. Targeting of kinases has been explored through different methods, including using small molecules, RNA interference and knockout cell lines.

- Yaron et al. first tested inhibition of SRPK using RNA interference. When SRPK1 was knocked down with siRNA, there was a decrease in N phosphorylation and viral RNA amounts. Next, they tested small molecule inhibitors of SRPK1 (SPHINX31 and SRPIN340) which resulted in decreased viral RNA and infectious viral titer in a dose-dependent manner. Finally, Yaron et al. tested an FDA-approved SRPK1/2 inhibitor, alectinib, which also resulted in decreased viral propagation and N protein phosphorylation.
- Liu et al. showed that blocking GSK-3 activity decreases viral replication in cells and reduced infection in patients. They used CRISPR/Cas9 to knock down or knockout both GSK3 α and GSK3 β . They showed the combined KO of GSK3 α and GSK3 β prevented N protein phosphorylation. They also tested blocking GSK3 activity using small molecules (CHIR99021 and Enzastaurin). They found that both molecules stopped the accumulation of phosphorylated N protein in infected cells and inhibited SARS-CoV-2 infection.

We referred to this in our discussion as our results support the exploration of targeting kinases as a potential treatment for COVID-19. Based on the number of groups that have already investigated this treatment avenue in cell culture, we do not believe that we can add to the

current literature. We hope that our study can provide mechanistic biophysical insights to support others interested in continuing the translational studies with these potential treatments.

1. Yaron, T. M. et al. Host protein kinases required for SARS-CoV-2 nucleocapsid phosphorylation and viral replication. *Sci. Signal.* 15, 1–16 (2022).
2. Liu, X. et al. Targeting the Coronavirus Nucleocapsid Protein through GSK-3 Inhibition. *medRxiv* 2021.02.17.21251933 (2021).

Minor

1. In Fig.2D, the authors compared the behavior of N and pN, why data from different timepoint settings are shown.

The timepoints selected reflect the relevant timeframes for each type of interaction. Assessing binding of a condensate to the membrane is fast, and therefore interactions where no binding occurs has a short timeframe. On the other hand, the process of engulfment of a condensate or relaxation at the surface takes several minutes. We included supplemental movies that depict this more clearly – for example, condensates that do not bind to the GUV surface within a few seconds can be held to the surface using optical tweezers for several minutes and then removed from the surface with no impact on its binding behavior (e.g., see Supplemental Movie 6).

2. N is known to regulate the innate immune response, does phosphorylated N play a role in this?

There is some evidence to support that phosphorylation of N can affect its function in regulating the immune response, though more investigation is needed.

Several groups have investigated how phosphorylation affects N's interaction with G3BP1, which is a key component of stress granules involved in the innate immune response. Wu et al. showed that mutations to S51, a phosphorylation site on N that is outside of the SR-rich region, weakened the interaction between N and G3BP1 and caused the formation of stress granules. Lu et al. investigated N vs. a version of N lacking the SR-rich region and showed that both proteins are able to interact with G3BP1 in cells.

1. Wu J, et al. A novel phosphorylation site in SARS-CoV-2 nucleocapsid regulates its RNA-binding capacity and phase separation in host cells. *J Mol Cell Biol*, (2022).
2. Lu, S. et al. The SARS-CoV-2 nucleocapsid phosphoprotein forms mutually exclusive condensates with RNA and the membrane-associated M protein. *Nat. Commun*, (2021).

Phosphorylation of N has been shown to allow for interaction with 14-3-3 proteins. Tugaeva et al. investigated the potential consequences of this on the immune system. They said: “Conclusively, 14-3-3 binding to the SR-rich region of N holds potential to regulate multiple host cell processes affected by N. 14-3-3 binding to pN may present a cell immune-like response to the viral infection aimed at arresting or neutralizing N activities. On the other hand, in light of the

abundance of N protein in the infected cell, pN may instead arrest 14-3-3 proteins in the cytoplasm and indirectly disrupt cellular processes involving 14-3-3. For example, 14-3-3 ϵ and 14-3-3 η each play a role in the innate immune response via RIG-1 and MDA5 signaling, respectively. The N protein:14-3-3 interaction would modulate these and other signaling pathways involving 14-3-3 proteins.”

3. Tugaeva KV, Hawkins DEDP, Smith JLR, Bayfield OW, Ker DS, Sysoev AA, Klychnikov OI, Antson AA, Sluchanko NN. The Mechanism of SARS-CoV-2 Nucleocapsid Protein Recognition by the Human 14-3-3 Proteins. *J Mol Biol.* 2021 Apr 16;433(8):166875. doi: 10.1016/j.jmb.2021.166875. Epub 2021 Feb 5. PMID: 33556408; PMCID: PMC7863765.

3. Several references are missing (PMIDs: 32901111, 33837182, 34239064).

We thank the reviewer for bringing this literature to our attention. We have now included these references (page 4, lines 81-82, in reference to N phase separating with RNA, both in vitro and in vivo).

Reviewer #3 (Remarks to the Author):

The paper by Favetta et al. focuses on the role of phosphorylation in the N protein of CoV-2. I was specifically asked to focus on the SAXS part of the paper.

However, I cannot help but notice that when N phosphorylation is discussed in the introduction, the interaction with 14-3-3 proteins that is phosphorylation dependent (14-3-3 make up 1% of all soluble proteins in a cell) is neither taken into account nor discussed. I consider this a significant drawback. Structures of N in complex with 14-3-3 exist and are well described in the literature and certainly N behaves differently when bound to 14-3-3.

We thank the reviewer for bringing this literature to our attention. We edited our discussion to include details on the limitations of our study, such as not considering many of the proteins N is known to interact with. We specifically highlight 14-3-3 proteins given that phosphorylation has been specifically identified as regulating that interaction (page 34, lines 720-723).

Concerning SAXS analysis, the Guinier plot (Figure S11) of N at 4 mg/ml reveals a little bit of aggregation, these data should be used with caution, or preferable not used at all.

We agree that the SAXS scattering curves for our proteins at 4 mg/mL indicate a small amount of aggregation. We removed the 4 mg /mL data from figure 6, B and C. The difference in radius of gyration from figure 6B becomes more significant with the removal of the 4 mg / mL data, while the difference in maximum dimension remains not significant.

What would really help the SAXS analysis it to calculate theoretical scattering curves based on computer simulations or the phosphorylated N and non-phosphorylated N (for non-

phosphorylated N this was already done). However, this would be a significant effort and while improving a lot Figure 5 section F-J it would not change the main message of the study, so my recommendation to the editors is not to insist on it.

We thank the reviewer for this suggestion. Given that we wanted to better understand the types of dynamic, intra-dimer interactions that were promoted by phosphorylation, we heeded this suggestion and worked with collaborators (now included as authors on the paper) to generate an ensemble from a coarse-grained simulation for N and a phosphomimetic version of N. We were able to reweigh our ensemble of conformations from the model based on our SAXS data to best represent physiological conformations of N and pN.

Our new results give us more information on which residues interact more frequently upon phosphorylation. We learned that with phosphomimetic mutations, each part of the N monomer within a dimer loses some types of interactions while gaining others, both with itself and with the second monomer within the N dimer. For example, the linker region of N interacts more with the opposite RNA binding domain, linker domain and dimerization domain. We believe these new interactions occupy the linker region and interfere with the protein's ability to interact with RNA. These results were included in Figure 6, copied below. Details on which residues are forming new interactions are included in Supplementary Figure S18.

Figure 6: N protein phosphorylation weakens RNA binding affinity due to change in protein conformation. A) Representative pairwise interatomic distance distribution $P(r)$ derived from SAXS for N and pN. N or pN concentration = 3 mg/mL. B) The average maximum distance

(D_{max}) and radius of gyration (R_g) for N and pN derived from the pair distance distributions. Error bars represent one standard deviation (± 1 s.d.). p values were determined using two-sided student's t-test; asterisk indicates $p < 0.05$. C) Normalized Kratky plot comparing the scattering of N vs. pN, indicating a structural change has occurred due to phosphorylation. Concentrations shown for N or pN are 1, 1.5, and 3 mg/mL from lighter to darker. D) Bead model representation for the N dimer developed from SAXS results (left) and hypothesized conformation of N (right). E) Bead model representation for the pN dimer developed from SAXS results (left) and hypothesized conformation of pN highlighting new intermolecular interactions (right). F) Snapshots from N simulation depicting conformations that demonstrate the protein's representative collapsed and extended states. The radius of gyration of each selected conformation is noted. G) Comparison of the pairwise interatomic distance distribution P(r) derived from SAXS and molecular dynamics simulation before and after reweighting with SAXS data for N. H) Comparison of the pairwise interatomic distance distribution P(r) derived from SAXS and molecular dynamics simulation before and after reweighting with SAXS data for pN. I) Change in contact probability between all residues within a dimer, compared between N and phosphomimetic N simulations. Lower left and upper right quadrants represent intramonomer interactions while the upper left and lower right quadrants represent intermonomer interactions. Blue coloring indicates more frequent interactions occurred in phosphomimetic N compared to unmodified. J) Analysis of the difference in intra-monomer (left) and inter-monomer (right) interactions between N and phosphomimetic N, per domain of the protein (as defined by the schematic of N above). Positive numbers / blue coloring indicates more frequent interactions occurred in phosphomimetic N compared to unmodified. Below, sum of change in interactions across each domain for both intra- and inter-domain interactions.

I also have few comments beyond SAXS. Major comments:

1) "GUVs were made with a lipid composition meant to approximate the human ER membrane, with 60% DOPC, 25% DOPE, 10% DOPS, and the addition of 5% Ni-NTA lipids." Seriously, a membrane without cholesterol was prepared to mimic ER membrane?

We thank the reviewer for bringing this to our attention. We had originally chosen to simplify the composition of our GUVs, and followed the example of others who did not include cholesterol in their GUVs mimicking the endoplasmic reticulum (ER) membrane [1-3]. Note that although the ER is the main organelle synthesizing cholesterol, the lipid is rapidly transported to other organelles such that it displays comparatively lower concentrations of sterols and complex sphingolipids; sources estimate the cholesterol content of the ER is ~ 5% [4-5].

However, upon further research, we found references of lipidomic studies on SARS-CoV-2 viral capsids that show that viral membranes are more similar to lysosome membranes, which is one of the known budding pathways. These membranes were found to have ~25% cholesterol [6].

We also recognize that even a small percent of cholesterol in a membrane can affect its properties. A recent publication investigated the effect of cholesterol on condensate interaction with GUVs and found that even with as little as 5% cholesterol, condensates tended to wet membranes to a lower degree [7].

Therefore, we repeated our GUV experiments in Figure 2 with GUVs of the following composition:

- + 2% cholesterol: 58% DOPC, 25% DOPE, 5% Ni-NTA, 10% DOPS, 2% cholesterol
- + 5% cholesterol: 55% DOPC, 25% DOPE, 5% Ni-NTA, 10% DOPS, 5% cholesterol
- + 25% cholesterol: 35% DOPC, 25% DOPE, 5% Ni-NTA, 10% DOPS, 25% cholesterol

The presence of cholesterol did not affect whether N or pN protein bound to GUVs, but inclusion of 5 or more percent cholesterol did change the degree of wetting of condensates when wetting occurred. At 2% cholesterol, we did not observe any changes in behavior of the protein interacting with the condensate. At 5% cholesterol, we observed a slight reduction in wetting of condensates. At 25% cholesterol, only partial wetting was observed.

We included this data in Supplementary Figure 3 which is copied below. Note that we opted to not include this data in the main text Figure 2, but we do discuss it in a new paragraph in the Results. Our main conclusion of these new experiments is that in the most biologically relevant scenarios, the results did not significantly change. For N-Nsp3 interaction in the ER, ~5% cholesterol is expected, where only minor changes in wetting occur. For N-M interaction in budding membranes, the binding and engulfment behavior did not change. Therefore, we thought it was clearer to keep a constant GUV composition in the main text.

Figure S3. Effect of cholesterol on N condensate interaction with membranes. A) Representative widefield images showing the interaction between condensates and membranes over time, with GUVs incorporating 2, 5, and 25% cholesterol, resulting in final lipid compositions of:

+ 2% cholesterol: 58% DOPC, 25% DOPE, 5% Ni-NTA, 10% DOPS, 2% cholesterol

+ 5% cholesterol: 55% DOPC, 25% DOPE, 5% Ni-NTA, 10% DOPS, 5% cholesterol

+ 25% cholesterol: 35% DOPC, 25% DOPE, 5% Ni-NTA, 10% DOPS, 25% cholesterol.

GUVs are labeled in green, and condensates in blue. Interaction type is qualitatively classified. Scale bars = 5 μ m. B) Quantification of the geometric factor from the interaction of pN and 1-1000 RNA condensates, with GUVs coated with NSP3¹⁻¹¹¹ and varying cholesterol compositions (data with 0% cholesterol from Figure 2). p values were determined using one-way ANOVA followed by post hoc Tukey's test. (NS, not significant; * p < 0.05; ** p < 0.01).

1. Chorlay, A. et al., Origin of gradients in lipid density and surface tension between connected lipid droplet and bilayer. *Biophysical Journal*, 24, 5491-5503 (2021).
2. Mathiassen, P.P.M., Menon, A.K. & Pomorski, T.G. Endoplasmic reticulum phospholipid scramblase activity revealed after protein reconstitution into giant unilamellar vesicles containing a photostable lipid reporter. *Sci Rep* 11, 14364 (2021).
3. Kovalev N, Pogany J, Nagy PD. Reconstitution of an RNA Virus Replicase in Artificial Giant Unilamellar Vesicles Supports Full Replication and Provides Protection for the Double-Stranded RNA Replication Intermediate. *J Virol*. 2020 Aug 31;94(18):e00267-20.
4. van Meer, G. Membrane lipids, where they are and how they behave: Sphingolipids on the move. *FASEB J*. 24, 112–124 (2010).
5. Casares D, Escribá PV, Rosselló CA. Membrane Lipid Composition: Effect on Membrane and Organelle Structure, Function and Compartmentalization and Therapeutic Avenues. *Int J Mol Sci*. 2019 May 1;20(9):2167
6. Saud Z, et al. The SARS-CoV2 envelope differs from host cells, exposes procoagulant lipids, and is disrupted in vivo by oral rinses. *J Lipid Res*. 2022 Jun;63(6):100208.
7. Mangiarotti, A., Sabri, E., Schmidt, K.V. et al. Lipid packing and cholesterol content regulate membrane wetting and remodeling by biomolecular condensates. *Nat Commun* 16, 2756 (2025)

2) “We performed experiments at two protein concentrations: low concentration, at which N is in its monomeric form, and high concentration, at which dimers form if the dimerization domain is present.” I believe that the experiments were done correctly. However, a little more explanation of how single molecule FRET was done in high concentration would help. Was the labeled protein mixed with unlabeled protein, or some other “trick” was used?

We apologize that our explanation was not sufficiently detailed. In our smFRET experiments, protein concentrations were (1) 100 pM labeled protein for the low concentrations or (2) 100 pM labeled protein + 1 μ M unlabeled protein for high concentration for the full-length construct or (3) 100 pM labeled protein + 4 μ M unlabeled protein for high concentration for N¹⁻²⁴⁶. (Here, the higher 4 μ M protein concentration was used to show that additional protein in the experiment would not affect the results since dimerization of the N fragment lacking the dimerization domain would not happen even at much higher protein concentrations). We have now included this information in the main text (page 25, lines 504– 506).

Minor comments:

1) “The N-terminal folded domain (NTD) strongly binds with specific viral RNA elements, stabilizing the RNP complex”. RNP should be defined when it is used for the first time.

Please note that ribonucleoprotein (RNP) complex is first used and defined in page 2, line 44.

2) “The central disordered region acts as a linker that allows for conformational flexibility. More specifically, the serine/arginine (SR)-rich region (aa 175 – 206; Figure 1B) within the linker is known to participate in both protein–protein and protein–RNA interactions.”

SAXS derived structures of N are available in the literature, they should be discussed here.

We thank the reviewer for the suggestion. We included information on how the N protein linker was found to be partially extended in solution, which suggests the existence of some residual structures, while permitting the protein conformational flexibility [1] (page 3, lines 56-58). We also added a sentence in the results section noting that our SAXS results for the unmodified N protein are in agreement with previously published SAXS data (page 24, lines 481-482).

1. Zeng, W. et al. Biochemical characterization of SARS-CoV-2 nucleocapsid protein. *Biochem. Biophys. Res. Commun.* 527, 618–623 (2020).